# Unifying Autoregressive and Diffusion-Based Sequence Generation

**Nima Fathi,**\* **Torsten Scholak & Pierre-André Noël**
ServiceNow Research
nima.fathi@mila.quebec, {torsten.scholak,pierre-andre.noel}@servicenow.com

## Abstract

We present significant extensions to diffusion-based sequence generation models, blurring the line with autoregressive language models. We introduce *hyperschedules*, which assign distinct noise schedules to individual token positions, generalizing both autoregressive models (*e.g.*, GPT) and conventional diffusion models (*e.g.*, SEDD, MDLM) as special cases. Second, we propose two *hybrid token-wise noising processes* that interpolate between absorbing and uniform processes, enabling the model to fix past mistakes, and we introduce a *novel inference algorithm* that leverages this new feature in a simplified context inspired from MDLM. To support efficient training and inference, we design attention masks compatible with KV-caching. Our methods achieve state-of-the-art perplexity and generate diverse, high-quality sequences across standard benchmarks, suggesting a promising path for autoregressive diffusion-based sequence generation. See code and resources at https://hdlm-colm.github.io/.

## 1 Introduction

Generative diffusion models, primarily recognized for their impressive image generation performance in continuous domains (Yang et al., 2023), are rapidly gaining traction in language modeling, a discrete domain historically dominated by autoregressive (AR) models such as the GPT family (Radford et al., 2019; Brown et al., 2020). Contrary to the apparent separation between AR and diffusion models, and despite their distinct historical development, this work reveals a fundamental connection: AR models are a form of diffusion.

The core principle behind diffusion models involves *prescribing* a "noising" process that gradually destroys information in training data samples, subsequently *learning* a neural network that progressively generates new samples from "pure noise" with a denoising process. The noising process acts as a form of data augmentation: together with the original training dataset, it specifies the *curriculum* on which the generator (denoiser) is trained. Part of the attraction for these models arises from their rich theoretical grounding, resulting in concrete practical techniques. In particular, a model's training and inference environments can be decoupled, allowing for a compute-budget knob at inference time, and guidance techniques adapting a model's behavior to specific situations.

Despite the common use of Gaussian noise in continuous diffusion, the underlying principles can be adapted to discrete state spaces (Austin et al., 2021; Zhou et al., 2023; Lou et al., 2024). Common practices for sequence generation have the noising process randomly and independently substituting some original tokens by completely unrelated ones, *i.e.*, *uniformly* sampled tokens or a special "mask" *absorbing* state. A noise schedule determines token replacement probabilities at different points in the curriculum. The resulting sequence at the schedule's highest noise level retains no mutual information with the original sequence. The generator is trained on this curriculum to enable the production of novel sequences.

This work unifies AR and diffusion sequence generation by introducing *hyperschedules*, allowing different positions in the sequence to be affected by different noise schedules. We

---

\*Also at Québec Artificial Intelligence Institute (Mila) and McGill University.

The training curriculum gradually destroys information from data sample (noising).

$$\mathbf{X}_T \xrightarrow{q_{T-1|T}} \mathbf{X}_{T-1} \xrightarrow{q_{T-2|T-1}} \cdots \xrightarrow{q_{1|2}} \mathbf{X}_1 \xrightarrow{q_{0|1}} \mathbf{X}_0$$

$$\hat{\mathbf{X}}_T \xleftarrow{p^{\boldsymbol{\theta}}_{T|T-1}} \hat{\mathbf{X}}_{T-1} \xleftarrow{p^{\boldsymbol{\theta}}_{T-1|T-2}} \cdots \xleftarrow{p^{\boldsymbol{\theta}}_{2|1}} \hat{\mathbf{X}}_1 \xleftarrow{p^{\boldsymbol{\theta}}_{1|0}} \hat{\mathbf{X}}_0$$

At inference, new samples are generated in $T$ generation steps (denoising).

Figure 1: Generative diffusion models *prescribe* (through $q_{t|t+1}$) a curriculum process $\{\mathbf{X}_t\}$, then *learn* (through $p^{\boldsymbol{\theta}}_{t+1|t}$) a reverse process $\{\hat{\mathbf{X}}_t\}$ so that the marginal distributions match at each step $t$ (vertical squiggly lines). $\mathbf{X}_T$ is the training dataset and $\hat{\mathbf{X}}_T$ is the generated output. This work focuses on discrete diffusion for sequences of discrete tokens. We show that standard autoregressive models (*e.g.*, GPT) are an extreme case of this framework, a unification enabling a vast continuum of diffusion models, including autoregressive ones.

establish that autoregressive models, such as GPT, can be understood as diffusion models without data augmentation, utilizing a discrete noise schedule comprising only "full noise" and "no noise" levels. This unification expands model design space and enables a variety of generalized AR-like approaches. With the hindsight, hyperschedule-like concepts can be identified in the "Mask" process of Bansal et al. (2023) as well as in the "Swin-DPM" technique of Feng et al. (2024).

Recent (Sahoo et al., 2024; Ou et al., 2024; Shi et al., 2024) and concurrent (Liu et al., 2024; Kim et al., 2025; Peng et al., 2025; Nie et al., 2025; Wang et al., 2025; Arriola et al., 2025) works have focused on mask diffusion models (MDMs). By specializing on the "absorb" noising mechanism, these MDMs enable great simplifications over a general-purpose treatment: neural networks no longer need an explicit noise-level dependency, and a more standard loss function can be used. However, our work shows that the same feats and simplifications can be accomplished in a general, non-MDM case. Motivated by the same rationale that led concurrent MDMs to conceive "remasking" strategies, we introduce hybrid noising processes, interpolating between the "absorb" and "uniform" processes to combine the benefits of both and achieve state-of-the-art performances, with aspects further improved by our novel adaptive correction sampler (ACS) inference algorithm. Our hyperschedule-equipped approach supports specialized attention matrices enabling KV-caching and efficient training.

In summary, our main contributions are:

- we unify AR and diffusion sequence generation by introducing *hyperschedules*;
- we consider hybrid noising processes, reaping benefits from both leading noising processes and achieving state-of-the-art performances, with and without our novel ACS inference algorithm; and
- our hyperschedule-powered hybrid processes generalize multiple concurrent developments to non-MDM setting, including efficient training and KV-caching.

## 2 Unification Through Abstraction

In this section, we reconcile autoregressive and diffusion-based language models by abstracting-out their respective implementation details, instead emphasizing their shared essence.

**Sequence Generator** We define a *sequence generator* as a stochastic procedure that repeatedly calls a neural network to yield a sequence of $d$ tokens, each taken from a finite vocabulary $\mathcal{X}$. We write $\hat{\mathbf{x}}_t = (\hat{x}^0_t, \cdots, \hat{x}^{d-1}_t) \in \mathcal{X}^d$ the *state* of the generator after $t$ calls: starting from a (trivial) initial state $\hat{\mathbf{x}}_0$, the neural network specifies the probability $p^{\boldsymbol{\theta}}_{t+1|t}(\hat{\mathbf{x}}_{t+1}|\hat{\mathbf{x}}_t)$ to get to state $\hat{\mathbf{x}}_{t+1}$ from state $\hat{\mathbf{x}}_t$. The generated sequence $\hat{\mathbf{x}}_T$ is obtained after $T$ such calls. The bottom row of Figure 1 illustrates this process (from right to left, writing $\hat{\mathbf{X}}_t$ the stochastic variable taking value $\hat{\mathbf{x}}_t$).

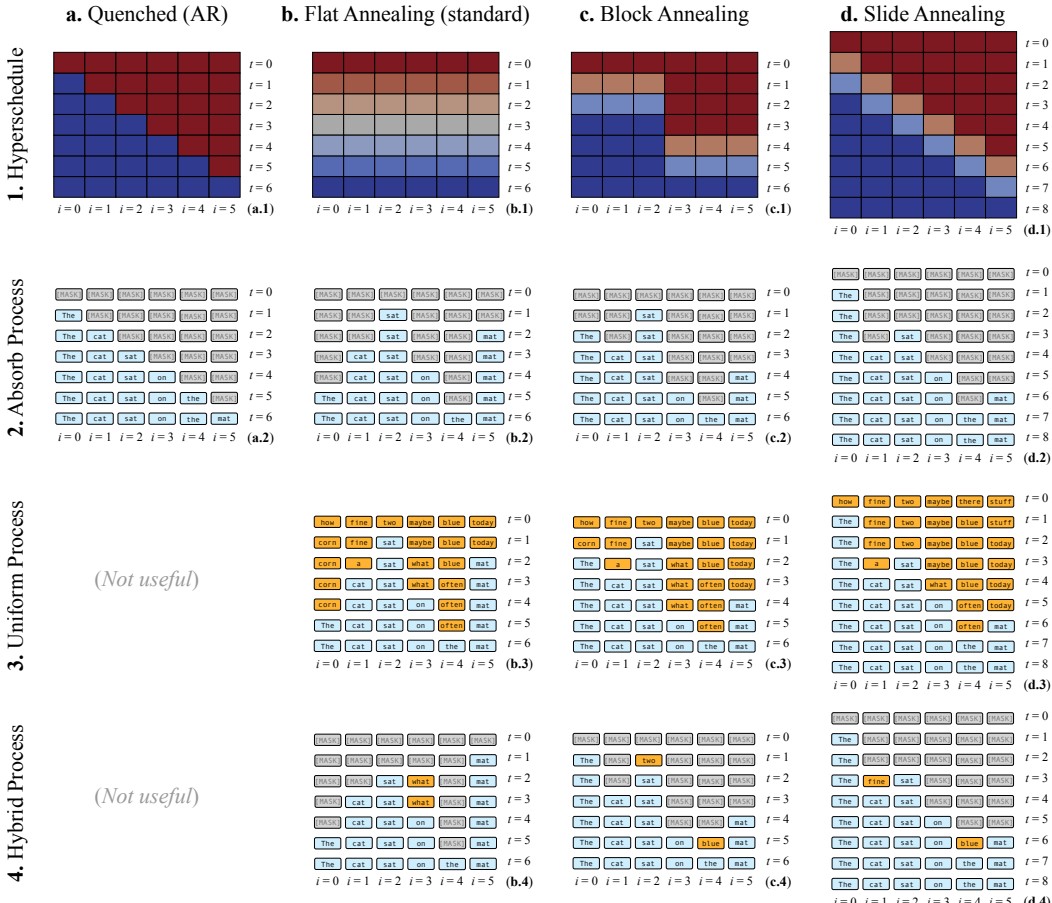

Figure 2: We introduce $\tau$-*hyperschedules* (top row) – subjecting different token positions $i$ with different noise levels (red high; blue low) at different generation step $t$ – and illustrate how three different noising processes (bottom 3 rows) can be modulated by such hyperschedules. **Hyperschedules.** (a) Standard AR models (*e.g.*, GPT) determine tokens one by one, "quenching" each of them to full determination in a single step. They may thus be construed as an extreme case of a diffusion model. (b) Standard diffusion models (*e.g.*, SEDD) gradually anneal all tokens independently of their position. (c) Block-wise application of flat annealing, here for blocks of width $\omega = 3$. (d) Annealing with a sliding window ("smoothed" AR), here using window width $\omega = 3$. These last two examples share important features of both AR and diffusion models. While the 4 presented examples all generate $\rho = 1$ token per step in the long-sequence limit (with the caveat that slide experiences an initial overhead of $\omega - 1$ steps), the last 3 patterns are all straightforwardly adapted to $\rho > 1$ ("quick draft") and $\rho < 1$ ("think hard") regimes. **Noising Processes.** (2) The *absorb* noising process – a.k.a. Masked Diffusion Model (MDM) – overwrites tokens with a special MASK token. These masks are "known unknowns": it is clear to a denoising model that it must put a non-mask token in its stead. Conversely, unmasked tokens are taken as "absolute truth" during generation: once a token has been unmasked, it remains unaltered until the end. (3) The *uniform* noising process overwrites with (non-mask) tokens selected uniformly at random from the vocabulary of possible tokens. The orange/light-blue color coding is not part of the state $\mathbf{x}_t$, and is provided solely for the reader's convenience. The model thus has no direct way to know if a token has been altered before ("unknown unknowns"), and may thus revisit a position's value many times during generation. (4) The *hybrid* noising process blends a little bit of *uniform* into the *absorb* process. When denoising, MASK tokens still represent clear "known unknowns", while unmasked ones have become "*a priori* good" candidates that may however need to be "fixed". We argue that it is desirable for models to learn to fix their own mistakes.

**Discrete Diffusion Sequence Generators**   A sequence generator's neural network must *learn* the function $p^{\theta}_{t+1|t}(\hat{\mathbf{x}}_{t+1}|\hat{\mathbf{x}}_t)$ such that the yielded $\hat{\mathbf{x}}_T$ approximates the distribution of clean training samples $\mathbf{x}_T$ (no hat!) from the training dataset. To this end, the "diffusion" paradigm (Yang et al., 2023) is to *prescribe* a distribution $q_{t-1|t}(\mathbf{x}_{t-1}|\mathbf{x}_t)$ from which we can sample $\mathbf{x}_{t-1}$ given $\mathbf{x}_t$. This *noising* process (top row in Figure 1) gradually destroys the information in the training data sample $\mathbf{x}_T$, until nothing remains of it in $\mathbf{x}_0$; we call *curriculum* the process $\{\mathbf{X}_t\}$ obtained by recursively sampling the prescribed $q_{t-1|t}(\mathbf{x}_{t-1}|\mathbf{x}_t)$ starting from a training sample $\mathbf{x}_T$. The training objective for $p^{\theta}_{t+1|t}$ amounts to *denoising* such noisy samples, aiming to align the marginal distribution of $\hat{\mathbf{X}}_t$ with its $\mathbf{X}_t$ counterpart (noted $\hat{\mathbf{X}}_t \rightsquigarrow \mathbf{X}_t$ in Figure 1).

**Token-Wise Processes**   The noising process is typically constructed out of many independent processes, each affecting one of the original tokens, *i.e.*, $q_{t|t+1}(\mathbf{x}_t|\mathbf{x}_{t+1}) = \prod_i q^i_{t|t+1}(x^i_t|x^i_{t+1})$. In practice, $q^i_{t|t+1}(x^i_t|x^i_{t+1})$ is given by the $x$-th column of a matrix $Q_{t|t+1}$ that is independent of the position $i$. For example, SEDD (Lou et al., 2024) uses $Q_{t|t+1} = \exp\big((\bar{\sigma}_{T-t} - \bar{\sigma}_{T-t-1})Q_{\text{tok}}\big)$, where the transition matrix $Q_{\text{tok}}$ is either of

$$Q_{\text{Uniform}} = \begin{bmatrix} \frac{2-|\mathcal{X}|}{|\mathcal{X}|-1} & \cdots & \frac{1}{|\mathcal{X}|-1} & 0 \\ \vdots & \ddots & \vdots & \vdots \\ \frac{1}{|\mathcal{X}|-1} & \cdots & \frac{2-|\mathcal{X}|}{|\mathcal{X}|-1} & 0 \\ 0 & \cdots & 0 & 0 \end{bmatrix} \qquad Q_{\text{Absorb}} = \begin{bmatrix} -1 & \cdots & 0 & 0 \\ \vdots & \ddots & \vdots & \vdots \\ 0 & \cdots & -1 & 0 \\ 1 & \cdots & 1 & 0 \end{bmatrix}, \qquad (1)$$

and $\bar{\sigma}_0 \leq \bar{\sigma}_1 \leq \cdots \leq \bar{\sigma}_T$ are cumulative noise schedules such that $\bar{\sigma}_0 \approx 0$ and $\bar{\sigma}_T \approx \infty$. Here $|\mathcal{X}|$ is the number of tokens in the set $\mathcal{X}$, including a special MASK token associated with the last dimension of $Q_{\text{tok}}$. In words, using $Q_{\text{Uniform}}$ gradually replaces tokens by random ones, while $Q_{\text{Absorb}}$ gradually replaces them by MASK. Generation starts from $\hat{\mathbf{x}}_0$ sampled from the stationary distribution $p_0$ (*i.e.*, random non-mask tokens for $Q_{\text{Uniform}}$ and all-masks for $Q_{\text{Absorb}}$), and $p^{\theta}_{t+1|t}$ is learned in terms of diffusion weighted denoising score entropy (DWDSE) (Lou et al., 2024).

Although MDMs technically correspond to the above for the case $Q_{\text{Absorb}}$, the literature has converged on a much simpler formulation in terms of $1 \approx \alpha_0 \geq \alpha_1 \geq \cdots \geq \alpha_T \approx 0$ such that $x^i_t$ has probability $\alpha_{T-t}$ to be the original token $y^i$ and probability $1 - \alpha_{T-t}$ to be MASK. The resulting transition matrix $Q_{t|t+1} = \mathbb{1} + (1 - \frac{\alpha_{T-t}}{\alpha_{T-t-1}})Q_{\text{Absorb}}$ may be further simplified, allowing training $p^{\theta}_{t+1|t}$ using a weighted cross-entropy loss (Sahoo et al., 2024).

**AR Sequence Generators**   Most modern language models predict tokens *autoregressively*: one token at a time, each conditional on the tokens that precede it. This may be viewed as a sequence generator with $T = d$: $\hat{\mathbf{x}}_0 = (\text{MASK}, \text{MASK}, \cdots, \text{MASK})$ represents an empty sequences using a special MASK token, and each $p^{\theta}_{t+1|t}(\hat{\mathbf{x}}_{t+1}|\hat{\mathbf{x}}_t)$ predicts the token $\hat{x}^t_{t+1}$ to substitute the first MASK encountered in $\hat{\mathbf{x}}_t$. This can be viewed as an extreme case of a diffusion model where the conditional probability $q_{t|t+1}$ is actually a deterministic function that masks-out the $t$-th token, ultimately resulting in $\mathbf{x}_0$ composed solely of MASK tokens. However, notice that this $q_{t|t+1}(\mathbf{x}_t|\mathbf{x}_{t+1}) = \prod_i q^i_{t|t+1}(x^i_t|x^i_{t+1})$ involves factors depending on the position $i$, which is forbidden in the discussion above; Section 3.1 palliates to this issue.

**Transformers**   All sequence generators considered in this work are implemented as transformers (Vaswani et al., 2017), more specifically the backbone from Peebles & Xie (2023). One subtlety is that many diffusion and/or masked language models use an ALIGNED configuration where each transformer cell predicts (output) the same token position as the one it receives (input). In contrast, AR models use the SHIFTED configuration: each cell predicts the *next* token in the sequence. These subtleties can be abstracted out at an high level (see details in Figure 4 in Appendix).

# 3 Autoregressive Sequence Diffusion

## 3.1 Hyperschedules: position-dependent schedules

We generalize standard diffusion curricula by subjecting different token positions $i$ to different noise schedules, distinguishing the number of generation steps $T$ from the number of noise levels $\mathcal{T}$. Concretely, $q^i_{t|t+1}(x'|x)$ is given by the $x$-th column of a $Q^i_{t|t+1}$ obtained by substituting each instance of $\sigma_{T-t}$ or $\alpha_{T-t}$ in $Q_{t|t+1}$ by $\sigma_{\tau^i_t}$ or $\alpha_{\tau^i_t}$, respectively, where the *hyperschedule* $\tau^i_t \in \{0, 1, \cdots, \mathcal{T}\}$ satisfying $\mathcal{T} = \tau^i_0 \geq \tau^i_1 \geq \cdots \geq \tau^i_T = 0$ for all positions $i \in \{0, \cdots, d-1\}$. In effect, the noise schedule ($\bar{\sigma}$ or $\boldsymbol{\alpha}$) unfolds differently at different positions; hyperschedules act as schedules for schedules. See examples on top row of Fig. 2.

We introduce two characterizations of an hyperschedule. First, we define the window width $\omega$ as the largest (among all steps $t$) number of positions $i$ for which $(\tau^i_t, \tau^i_{t+1})$ is neither $(0, 0)$ nor $(\mathcal{T}, \mathcal{T})$. All other things being equal, a lower $\omega$ offers more opportunities to improve inference time (see Sec. 3.3 for examples). Standard AR models use $\boldsymbol{\tau}_{\text{Quench}}$ with value 1 where $i \geq t$ and 0 elsewhere, and thus have $\omega = 1$ by construction (Fig. 2(a.1)). Standard diffusion models use $\boldsymbol{\tau}_{\text{Flat}}$ with value $\mathcal{T} - t$ (often called "noise level") for all $i$, using $\mathcal{T} = T$ and $\omega = d$ (Fig. 2(b.1)). Two $\omega$-parametrized examples are also provided: concurrent work on block diffusion (Arriola et al., 2025) may be understood in terms of $\boldsymbol{\tau}^\omega_{\text{Block}}$ (Fig. 2(c.1)), and we introduce novel $\boldsymbol{\tau}^\omega_{\text{Slide}}$ (Fig. 2(d.1)).

Second, we define the token generation rate $\rho$ as the long-sequence limit of the ratio $d/T$. All hyperschedules explicitly presented in Fig. 2 share the same $\rho = 1$: in the long run, they require one model call to generate one token. However, all but $\boldsymbol{\tau}_{\text{Quench}}$ may be readily adapted to "quick draft" (*i.e.*, $\rho > 1$) or "think hard" (*i.e.*, $\rho < 1$) regimes.

## 3.2 Hybrid Processes: training denoisers that can fix their mistakes

In the *absorb* process, each step produces the curriculum by replacing some of $\mathbf{x}_T$'s entries by the special MASK token. Conversely, at generation time, the only action available to the generator is to replace some MASK tokens by non-MASK ones. Notice that, unless the generator is "perfect", it may become apparent late in the generation process that some early token choices were, in retrospect, inherently incompatible. However, there are no action available for the model to "fix" these token choices: no backsies. Although such "hindsight" situations may occur in any domain, they are particularly relevant to computer code generation and other reasoning-intensive tasks.[1]

Conversely, the *uniform* process can replace any (non-MASK) token by any other one, both at curriculum specification and sequence generation. Thus, at no point in the generation process does the model have any indication whether it has already altered a given token before. We hypothesize that this may cause a "lack of commitment" on the model's part: how much should you "trust" the value of a token? At least with absorb, MASK tokens capture known unknowns.

These observations motivate an *hybrid* forward process, of which we consider two varieties. The first one uses the SEDD framework with $Q_{\text{tok}}$ set to $Q^\gamma_{\text{Hybrid}} = (1-\gamma)Q_{\text{Absorb}} + \gamma Q_{\text{Uniform}}$, interpolating the $Q_{\text{Uniform}}$ and $Q_{\text{Absorb}}$ extremes according to an hyperparameter $0 < \gamma < 1$. The evolution operator $\exp\big((\bar{\sigma}_{\tau^i_t} - \bar{\sigma}_{\tau^i_{t+1}})Q^\gamma_{\text{Hybrid}}\big)$ can be solved analytically (because $Q_{\text{Uniform}}$ and $Q_{\text{Absorb}}$ commute, see Appendix A), enabling use in practice with the standard SEDD loss.

Our second hybrid process variety is closer to the MDM framework: unless $\tau^i_t = 0$ (in which case $x^i_t = x^i_T$), the probability distribution for $x^i_t$ is given by the $x^i_T$-th column of $\big(\mathbb{1} + \epsilon Q_{\text{Uniform}}\big)\big(\mathbb{1} + (1 - \alpha_{\tau^i_t})Q_{\text{Absorb}}\big)$, where $0 < \epsilon < 1$ is a step-independent probability that the token is substituted by a uniform one, followed by a standard MDM process

---

[1]As an extreme example, consider the graph coloring of a particularly nasty instance.

henceforth. A weighted cross-entropy loss is used (see Appendix B.2) and, like MDMs, the neural network does not require an explicit noise-level dependency. Which of the two variety is used can be inferred from which of $\gamma$ or $\epsilon$ is specified.

### 3.3 Attention Mask and Efficiency

In SEDD, each call to the transformer predicts each entry of $\hat{x}_{t+1}$ in view of all entries in $\hat{x}_t$. In contrast, standard autoregressive models use a causal attention mask to ensure that $\hat{x}_{t+1}^t$ may only depend on $\hat{x}_t^{:t}$. Combined with the fact that $\hat{x}_t^{:t} = \hat{x}_T^{:t}$ at all step $t$, this causal maskenables inference-time efficiency improvements such as KV-caching. MDMs such as Sahoo et al. (2024) can enable a similar form of caching by relying on the special role of MASK tokens.

However, new opportunities for optimization come up when the hyperschedule follows a certain autoregressive-like regular structure such as the ones seen in Figs. 2(c.1, d.1). Indeed, at each steps $t$ the hyperschedule $\boldsymbol{\tau}_t$ and the state $\hat{x}_t$ both break in three components

$$\boldsymbol{\tau}_t = \boldsymbol{\tau}_t^{\text{settled}} \frown \boldsymbol{\tau}_t^{\text{active}} \frown \boldsymbol{\tau}_t^{\text{worthless}} \qquad \hat{x}_t = \hat{x}_t^{\text{settled}} \frown \hat{x}_t^{\text{active}} \frown \hat{x}_t^{\text{worthless}} , \qquad (2)$$

where: $\boldsymbol{\tau}_t^{\text{settled}}$ is composed exclusively of zeros and $\hat{x}_t^{\text{settled}}$ matches the first entries of $\hat{x}_T = \hat{y}$; both $\boldsymbol{\tau}_t^{\text{active}}$ and $\hat{x}_t^{\text{active}}$ have at most $\omega \ll d$ entries; and $\boldsymbol{\tau}_t^{\text{worthless}}$ is composed exclusively of repeated $\mathcal{T}$ while $\hat{x}_t^{\text{worthless}}$ bears no information about $\hat{y}$. Thus, when using an autoregressive attention mask on $\hat{x}_t^{\text{settled}}$, all the conditions are met to use KV-caching on these tokens just as in a standard autoregressive model. We may completely ignore $\hat{x}_t^{\text{worthless}}$, which leaves a small number $\omega$ of positions that densely attend to $\hat{x}_t^{\text{settled}} \frown \hat{x}_t^{\text{active}}$ when generating $\hat{x}_{t+1}^{\text{active}}$. See details in Appendix B.

### 3.4 Adaptive Correction Sampler

In addition to the theoretically-grounded inference schemes from SEDD and MDLM, we introduce *adaptive correction sampler* (ACS), a novel variation on MDLM's sampler that allows the model to alter the value of already-unmasked tokens, and that has empirically shown to perform particularly well for our hybrid process of the $\epsilon$-variety. We write $p_{\text{transfer}}^i$ the probability that MDLM's sampler (adapted to use our hyperschedule) would unmask the $i$-th token if it is masked, and proceed as usual for the token that are so masked. However, where MDLM would leave already-unmasked tokens as they are, ACS has probability $\eta(1 - p_{\text{transfer}}^i)$ to sample a replacement token from the model's prediction. $\eta$ serves as a hyperparameter modulating the intensity of this correction. Pseudocode for the original sampler and ACS sampler can be found in Appendix D.

## 4 Experiments

### 4.1 Experimental Setup

Our experiments investigate three primary design dimensions: (i) the choice of hyperschedule (*e.g.*, $\boldsymbol{\tau}_{\text{Flat}}$, $\boldsymbol{\tau}_{\text{Slide}}$, and $\boldsymbol{\tau}_{\text{Block}}$) illustrated in Fig. 2; (ii) transformer configurations, specifically ALIGNED and SHIFTED as depicted in Fig. 4; and (iii) the hybrid process, particularly our $\gamma$-Hybrid with $Q_{\text{Hybrid}}^{\gamma}$ operator and $\epsilon$-Hybrid parametrized by $\epsilon$. Additional implementation details are provided in Appendix B.

### 4.2 Language Model Likelihood Evaluation

We evaluate model likelihood performance across two established datasets.

**OPENWEBTEXT (OWT) (Gokaslan et al., 2019)**: Given that OWT lacks a standard data split, we follow Ou et al. (2024), reserving the last 100K documents as a held-out set for test perplexity reporting. All OWT models utilize a context length of 1024 tokens with the

| Mark | Method | Test PPL $\downarrow$ | $\gamma$ | $\tau$ |
|------|--------|--------|--------|--------|
| **(a)** | Baseline SEDD-Absorb [14] | 24.10 | 0 | $\tau_{\text{Flat}}$ |
| **(b)** | **(a)** $+$ Hybrid Process | 22.30 | 0.01 | $\tau_{\text{Flat}}$ |
| **(c)** | **(b)** $+$ Weighted token-embedding($= \gamma$-Hybrid) | 22.18 | 0.01 | $\tau_{\text{Flat}}$ |
| **(d)** | **(c)** $-$ Transformer time-conditioning | 22.47 | 0.01 | $\tau_{\text{Flat}}$ |
| **(e)** | **(c)** $+ \tau_{\text{Slide}}^{\omega=d}$ | 21.53 | 0.01 | $\tau_{\text{Slide}}^{\omega=d}$ |

Table 1: *Test Perplexity* for various design choices (lower is better), measured on the heldout 100k sample from **OWT** dataset. All ablations use ALIGNED with $d = 1024$.

| | | Parameters | PPL ($\downarrow$) |
|---|---|---|---|
| *Autoregressive* | OmniNet$_T$ (Tay et al., 2021) | 100M | 21.5 |
| | Transformer (65B tokens) (Sahoo et al., 2024)$^{\ddagger}$ | 110M | 22.3 |
| *Diffusion* | SEDD (65B tokens) (Lou et al., 2024) | 110M | 32.8 |
| | MDLM (65B tokens) (Sahoo et al., 2024) | 110M | 31.8 |
| | BD3-LMs $L' = 4$ (65B tokens) (Arriola et al., 2025) | 110M | 28.2 |
| *Diffusion (Ours)* | $\gamma$-Hybrid $_{[\gamma=0.02,\,\tau_{\text{Flat}},\,\text{ALIGNED}]}$ (56B tokens) | 110M | **27.8** |
| | $\gamma$-Hybrid $_{[\gamma=0.02,\,\tau_{\text{Flat}},\,\text{SHIFTED}]}$ (56B tokens) | 110 M | 28.3 |
| | $\gamma$-Hybrid $_{[\gamma=0.02,\,\tau_{\text{Block}}^{\omega=d/64},\,\text{ALIGNED}]}$ (65B) | 110M | **27.1** |
| | $\gamma$-Hybrid $_{[\gamma=0.02,\,\tau_{\text{Block}}^{\omega=d/4},\,\text{ALIGNED}]}$ (65B) | 110M | **27.0** |
| | $\gamma$-Hybrid $_{[\gamma=0.02,\,\tau_{\text{Block}}^{\omega=d/64},\,\text{SHIFTED}]}$ (65B) | 110M | **27.5** |
| | $\gamma$-Hybrid $_{[\gamma=0.02,\,\tau_{\text{Block}}^{\omega=d/4},\,\text{SHIFTED}]}$ (65B) | 110M | **26.6** |

Table 2: Test perplexities (PPL; $\downarrow$) on LM1B. Perplexity values for diffusion models are upper-bound estimations. $^{\dagger}$Reported in He et al. (2022). $^{\ddagger}$Reported in Sahoo et al. (2024). Best diffusion value is bolded.

GPT-2 tokenizer (Radford et al., 2019).

**LM1B (Chelba et al., 2014)**: We use the `bert-base-uncased` tokenizer, reporting test perplexity on the predefined test set. All LM1B models are trained with 128 tokens contexts.

### 4.2.1 Ablation Study

We conduct a systematic ablation study to assess the contributions of key design choices on OWT (Table 1). Our baseline model **(a)** follows the original SEDD-Absorb setup of Lou et al. (2024). Introducing our hybrid transition operator $Q_{\text{Hybrid}}^{\gamma}$ significantly improves performance, noted in configuration **(b)**. We further incorporate weighted token embeddings inspired by Ou et al. (2024), yielding additional improvement (configuration **(c)**; details in Appendix B.5). However, removing timestep conditioning (an operation that isn't theoretically justified for $\gamma$-Hybrid, in contrast with MDLM and $\epsilon$-Hybrid) slightly degraded performances (configuration **(d)**), prompting us to retain this component. We also explored the $\tau_{\text{slide}}^{\omega=1024}$ curriculum, which demonstrated modest improvements at the cost of increased computational overhead due to additional forward passes.

### 4.2.2 Language Modeling Performance

Following our ablation insights, we compare our best-performing configurations against state-of-the-art autoregressive and diffusion baselines on LM1B. Results presented in Table 2 illustrate significant improvement upon prior discrete diffusion baselines, reducing perplexity by approximately 19% relative to SEDD (Lou et al., 2024) and 3% compared to the strongest existing diffusion models.

| Method | PTB | WikiText | LM1B | Lambada | AG News | Pubmed | Arxiv |
|--------|-----|----------|------|---------|---------|--------|-------|
| Transformer 
(Sahoo et al., 2024) | 82.05 | 25.75 | 51.25 | 51.28 | 52.09 | 49.01 | 41.73 |
| SEDD Absorb‡ 
(Lou et al., 2024) | 96.33 | 35.98 | 68.14 | 48.93 | 67.82 | 45.39 | 40.03 |
| MDLM‡ 
(Sahoo et al., 2024) | 90.96 | 33.22 | 64.94 | 48.29 | 62.78 | 43.13 | 37.89 |
| BD3-LM $L' = 4$ 
(Arriola et al., 2025) | 96.81 | 31.31 | **60.88** | 50.03 | **61.67** | 42.52 | 39.20 |
| $\gamma$-Hybrid (444B) 
[$\gamma = 0.01$, $\tau_{\text{Flat}}$, ALIGNED] | **89.94** | **30.02** | 61.01 | **45.38** | 67.51 | 46.57 | 40.62 |
| $\epsilon$-Hybrid (444B) 
[$\epsilon = 0.01$, $\tau_{\text{Flat}}$, ALIGNED] | 90.89 | 32.53 | 68.91 | 50.23 | 64.61 | **41.18** | 37.85 |
| $\gamma$-Hybrid 
[$\gamma = 0.01$, $\tau_{\text{Slide}}^{\omega=d/4}$, ALIGNED] | 90.67 | 31.73 | 73.71 | **50.03** | 68.27 | **41.49** | 37.89 |
| $\gamma$-Hybrid 
[$\gamma = 0.01$, $\tau_{\text{Block}}^{\omega=d/64}$, SHIFTED] | 95.22 | 32.64 | 63.68 | **44.75** | 62.18 | **42.01** | 37.33 |

Table 3: Zero-shot unconditional perplexity on seven benchmark datasets from Lou et al. (2024) and Sahoo et al. (2024) and Arriola et al. (2025). ‡Reported in Arriola et al. (2025). All models are trained for 524B tokens unless otherwise stated. All diffusion models are upper bounds; the best diffusion value is **bolded**. See Appendix F.1 for complete results.

### 4.3 Zero-Shot Generalization

We evaluate the generalization capabilities of our models across seven standard datasets considered before by Lou et al. (2024) and Sahoo et al. (2024). As shown in Table 3, our Hybrid diffusion language family of models, ($\gamma$-Hybrid and $\epsilon$-Hybrid) consistently surpasses prior discrete diffusion approaches on 5 of the 7 benchmarks. Moreover, we narrow the gap between diffusion-based and AR models, outperforming AR baselines on two datasets.

### 4.4 Sampler Performance

We evaluate the performance gained from our proposed adaptive correction sampler (ACS). Table 4 empirically shows that ACS improves the $\epsilon$-Hybrid models' ability to correct initial missteps, resulting in sequences that are both more coherent and of higher quality. Pseudocode for the original sampler and ACS can be found in Appendix D.

| **Model Family** | **Sampler** | **MAUVE** (↑) | | **Gen PPL.** (↓) | | **Entropy** (↑) | |
|------------------|-------------|-----------|-----------|-----------|-----------|-----------|-----------|
| | | $\rho = 2$ | $\rho = 1$ | $\rho = 2$ | $\rho = 1$ | $\rho = 2$ | $\rho = 1$ |
| $\epsilon$-Hybrid 
[$\epsilon = 0.01$, $\tau_{\text{Flat}}$, ALIGNED] | Original Sampler 
 ACS | 0.848 
 **0.957** | 0.779 
 **0.947** | 121.90 
 **61.35** | 129.52 
 **43.98** | **5.49** 
 5.28 | **5.50** 
 5.18 |
| $\epsilon$-Hybrid 
[$\epsilon = 0.01$, $\tau_{\text{Block}}^{\omega=d/4}$, ALIGNED] | Original Sampler 
 ACS | 0.778 
 **0.813** | 0.847 
 **0.916** | 139.64 
 **71.77** | 142.13 
 **59.15** | **5.43** 
 5.38 | **5.46** 
 5.25 |

Table 4: Comparing the original sampler with ACS on $\epsilon$-Hybrid variants. For a more complete comparison please refer to Table 10.

### 4.5 Sequence Generation Quality-Diversity Trade-offs

We further investigate the trade-off between sequence generation quality and diversity by analyzing generative perplexity against token-level entropy and MAUVE[2] scores (Pillutla et al., 2021), as visualized in Fig. 3. Our hybrid configurations consistently achieve superior positions on both Pareto frontiers, indicating enhanced generation fluency, coherence, and diversity compared to baselines; a new state-of-the-art in diffusion language modeling.

---

[2]MAUVE strongly correlates with human judgment of text quality and diversity.

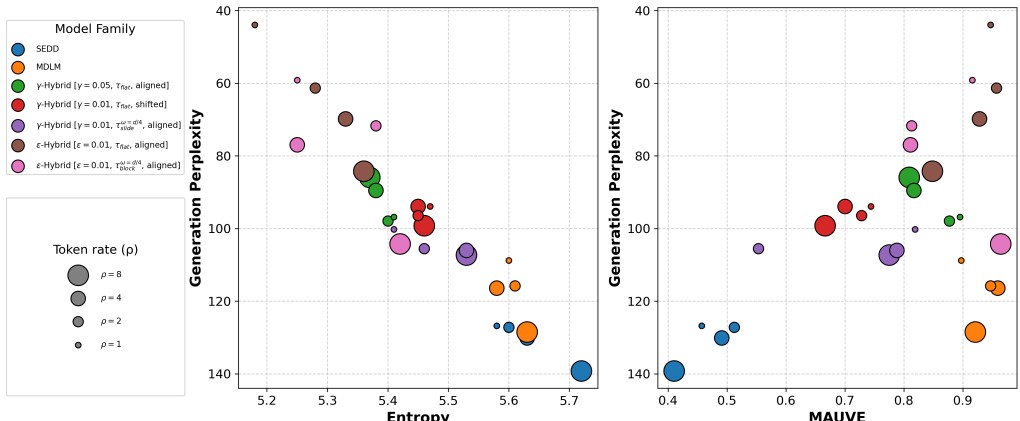

Figure 3: Left: Generative perplexity as a function of token-level entropy. Right: Generative perplexity versus MAUVE score. Our models consistently outperform baselines, achieving lower perplexity at comparable levels of diversity and fluency.

## 4.6 Additional Analyses and Results

Qualitative generation samples, both conditional and unconditional, are included in Appendix G. Hybrid models exhibit robust performance, effectively handling extended context lengths beyond the training range, particularly for OWT. Comprehensive ablations detailed in Appendices 4.2.1, F, and F.4 confirm the effectiveness of our design choices. Specifically, we find optimal performance around small $\gamma$ values (0.01 to 0.1), critical for balancing token commitment and flexibility, as evidenced in Table 8 and Fig. 10. Additionally, adjustments in generation rate $\rho$ notably impact both quality and inference speed (Table 9), with our proposed $\tau^{\omega}_{\text{Block}}$ schedules significantly benefiting from KV-caching (Table 11).

## 5   Conclusion

Diffusion-based language models offer some unique opportunities – including theory-supported guidance strategies and the native ability to iteratively improve their answer – but these benefits are no substitutes for raw language modeling performances. Staggering resources are continuously spent in scaling up AR models, engineering tools and techniques specialized to the AR paradigm. How could diffusion models even dream of catching up?

This work takes significant step toward a bold strategy: starting from an already-great AR language model, we wish to convert it (*e.g.*, fine-tuning) into an even better diffusion-based sequence generation model. This plan demands a SHIFTED configuration, an hyperschedules generalizing the AR concept (*e.g.*, $\tau_{\text{Slide}}$ or $\tau_{\text{Block}}$), and a curriculum (such as our hybrids) teaching the model how to generate quality sequences without painting itself into a corner.

Much of the design space opened by our innovations remain to be explored by future work. We have merely glanced at the realm of possible hyperschedules, and our success with $\epsilon$-Hybrid illustrates that more involved curricula can pay off, without the need to provide explicit noise levels to an eventual pre-trained AR model.

Our innovations also open the path for more fundamental work. Indeed, the limit $1 < \omega \ll d$ presents opportunities for tractable approximations of the joint distribution over $\omega$ tokens. On a different front, while our current approach employs a uniform distribution for replacing tokens, further improvements in diversity and quality may be achieved with distributions that more accurately reflect plausible, "honest" and/or "on policy" errors (rather than purely random token substitutions).

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

## A   Evolution Operators

Although our SEDD-based curricula are *a priori* defined in terms of an arbitrary $Q_{\text{tok}}$, actually using a model in practice demands that we can analytically solve the evolution operator $\exp(\Delta Q_{\text{tok}})$ for $\Delta \in \mathbb{R}^+$. This section re-derives the solutions for $Q_{\text{Uniform}}$ and $Q_{\text{Absorb}}$, then extends the results to $Q_{\text{Hybrid}}^\gamma$.

Using the definitions in Eq. (1), we can verify

$$(Q_{\text{Absorb}})^2 = -Q_{\text{Absorb}} \tag{3a}$$

$$(Q_{\text{Uniform}})^2 = -Q_{\text{Uniform}} \tag{3b}$$

$$Q_{\text{Uniform}} Q_{\text{Absorb}} = -Q_{\text{Uniform}} \tag{3c}$$

$$Q_{\text{Absorb}} Q_{\text{Uniform}} = -Q_{\text{Uniform}} \quad . \tag{3d}$$

Notice that, for any matrix $Q_*$ such that $(Q_*)^2 = \lambda Q_*$, we have

$$\mathrm{e}^{\phi Q_*} = \sum_{k=0}^\infty \frac{(\phi Q_*)^k}{k!} = \mathbb{1} + \lambda^{-1} Q_* \sum_{k=1}^\infty \frac{(\lambda\phi)^k}{k!} = \mathbb{1} + \lambda^{-1} Q_* \left[ -1 + \sum_{k=0}^\infty \frac{(\lambda\phi)^k}{k!} \right]$$
$$= \mathbb{1} - \lambda^{-1}(1 - \mathrm{e}^{\lambda\phi}) Q_* \quad . \tag{4}$$

Together with Eq. (3), we re-obtain the evolution operators used in SEDD

$$\mathrm{e}^{\Delta Q_{\text{Absorb}}} = \mathbb{1} + (1 - \mathrm{e}^{-\Delta}) Q_{\text{Absorb}} \tag{5a}$$

$$\mathrm{e}^{\Delta Q_{\text{Uniform}}} = \mathbb{1} + (1 - \mathrm{e}^{-\Delta}) Q_{\text{Uniform}} \quad . \tag{5b}$$

Now notice that $Q_{\text{Absorb}}$ and $Q_{\text{Uniform}}$ commute

$$[Q_{\text{Absorb}}, Q_{\text{Uniform}}] = Q_{\text{Absorb}} Q_{\text{Uniform}} - Q_{\text{Uniform}} Q_{\text{Absorb}} = 0 \quad , \tag{6}$$

which enables the analytical solution for $Q_{\text{Hybrid}}^\gamma$

$$\mathrm{e}^{\Delta Q_{\text{Hybrid}}^\gamma} \tag{7a}$$

$$= \mathrm{e}^{\Delta((1-\gamma) Q_{\text{Absorb}} + \gamma Q_{\text{Uniform}})} \tag{7b}$$

$$= \mathrm{e}^{(1-\gamma)\Delta Q_{\text{Absorb}}} \mathrm{e}^{\gamma\Delta Q_{\text{Uniform}}} \tag{7c}$$

$$= \left[ \mathbb{1} + (1 - \mathrm{e}^{-(1-\gamma)\Delta}) Q_{\text{Absorb}} \right] \left[ \mathbb{1} + (1 - \mathrm{e}^{-\gamma\Delta}) Q_{\text{Uniform}} \right] \tag{7d}$$

$$= \mathbb{1} + (1 - \mathrm{e}^{-(1-\gamma)\Delta}) Q_{\text{Absorb}} + (1 - \mathrm{e}^{-\gamma\Delta}) Q_{\text{Uniform}} + (1 - \mathrm{e}^{-(1-\gamma)\Delta})(1 - \mathrm{e}^{-\gamma\Delta}) Q_{\text{Absorb}} Q_{\text{Uniform}} \tag{7e}$$

$$= \mathbb{1} + (1 - \mathrm{e}^{-(1-\gamma)\Delta}) Q_{\text{Absorb}} + (1 - \mathrm{e}^{-\gamma\Delta}) Q_{\text{Uniform}} - (1 - \mathrm{e}^{-(1-\gamma)\Delta})(1 - \mathrm{e}^{-\gamma\Delta}) Q_{\text{Uniform}} \tag{7f}$$

$$= \mathbb{1} + (1 - \mathrm{e}^{-(1-\gamma)\Delta}) Q_{\text{Absorb}} + (\mathrm{e}^{-(1-\gamma)\Delta} - \mathrm{e}^{-\Delta}) Q_{\text{Uniform}} \quad . \tag{7g}$$

Equation (7g) is the desired analytical solution for the evolution operator.

**Additional notes on ELBO and hyperschedules.**   Theorem 3.6 from Lou et al. (2024) provides an ELBO bound enabling training with the DWDSE training loss. This Theorem holds for arbitrary constant-coefficients diffusion matrix, but is derived for the special case of position-independent schedules. One might thus wonder if the result still holds if we simply substitute in our position-dependent hyperschedules.

As mentioned in the beginning of Section 3.3 of Lou et al. (2024), such a general diffusion matrix would be exponential in size, justifying decomposing the process into $d$ independent subspaces in practical applications. The evolution operator for each of these subspaces has the form $\exp(\Delta Q_{\text{tok}})$ and, because of their independence, the full evolution operator is simply the product of their action on the independent subspaces.

When substituting in an hyperschedule instead of a schedule in SEDD's framework, the only difference is that the evolution operator for each subspace, $\exp(\Delta_i Q_{\text{tok}})$, gains a dependency

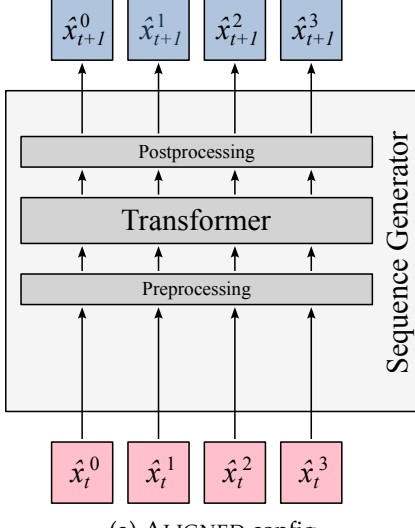

(a) ALIGNED config.

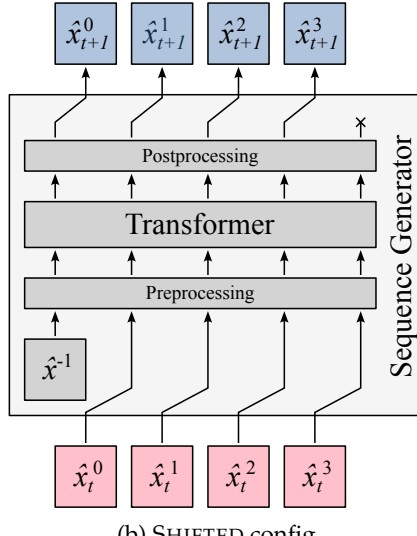

(b) SHIFTED config.

Figure 4: Two transformer-based sequence generators for $d = 4$. (a) The ALIGNED configuration of standard diffusion models is reminiscent of masked language models. (b) The SHIFTED configuration is closer to autoregressive language models. Here $\hat{x}^{-1}$ represent a token solely part of the conditioning (*i.e.*, not generated), and may or may not be constant (*e.g.*, BOS). Similarly, ⊀ represents that the output associated with the last token is discarded. Our position-based indexing abstracts away these details.

on the position *i* associated with this subspace. Crucially, the subspaces are still independent, and the overall evolution operator is still a simple product of the independent solutions. *All other derivations leading to SEDD's ELBO bound thus directly generalize to hyperschedules, including the case of $\gamma$-Hybrid.*

The case of $\epsilon$-Hybrid is more complicated. When $\epsilon = 0$, we fall back on MDLM, and the ELBO derivations from Sahoo et al. (2024) thus apply. However, for $\epsilon > 0$, the evolution operator cannot be represented under the form $\exp(\Delta Q_{\text{tok}})$: *we do not have an ELBO bound for $\epsilon$-Hybrid when $\epsilon > 0$.* In any case, our empirical observations support its use in practice.

It would be interesting to see if training with a proper ELBO-bounded loss could improve the performance of $\epsilon$-Hybrid models, and if something akin to our ACS algorithm could be derived from first principles. However, these concerns are outside the scope of the present work.

# B  Implementation Nuances

This section discusses several implementation details that affect both our model training and evaluation procedures.

## B.1  Training Setup

We now detail our experimental training procedure, structured into two distinct stages. In Stage 1, we initially train our base models in both ALIGNED and SHIFTED configurations using the hybrid noising process paired with the flat hyperschedule $\tau_{\text{flat}}$. Specifically, we adopt different modeling strategies depending on the chosen variant. For $\gamma$-variant models, we extend the existing discrete diffusion framework from Lou et al. (2024), integrating our proposed hybrid transition operator $Q^{\gamma}_{\text{Hybrid}}$. For $\epsilon$-variant models, we instead employ the newly proposed hybrid diffusion cross-entropy (HDCE) loss, detailed in Equation 8. Stage 1 models are trained separately on two standard datasets: *OpenWebText* and *LM1B*. Each configuration undergoes training for approximately 850K gradient updates with a batch

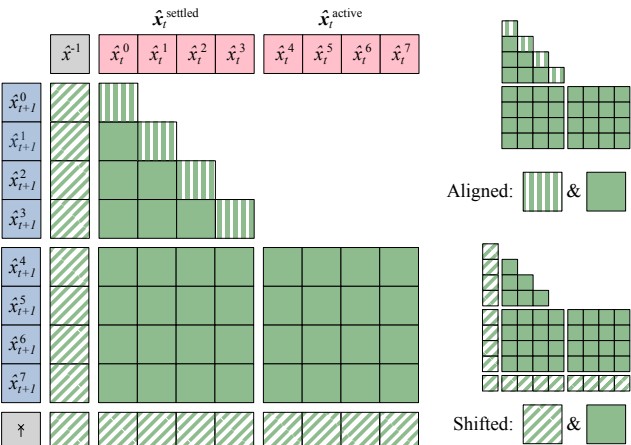

Figure 5: Example of attention mask for ALIGNED and SHIFTED configurations. Although these naive masks are appropriate for inference, directly training on them would be ineffi-cient; see Figures 6–9 for training-ready masks examples.

size of 512, processing roughly 444B tokens for OpenWebText and 56B tokens for LM1B. This pretraining took about 14 days on 8×A100-80 GB GPUs for OpenWebText and 7 days on 2×A100-80 GB GPUs for LM1B.

In Stage 2, we fine-tune the Stage 1 models under alternative hyperschedules—specifically, $\tau_{\text{block}}$ and $\tau_{\text{slide}}$. These experiments involve custom-designed attention masks tailored to each hyperschedule. Due to this customization, highly optimized attention kernels such as Flash-Attention are not applicable, necessitating reliance on the standard PyTorch attention mechanism, which incurs higher computational costs. Stage 2 training continues for an additional 150K gradient steps, resulting in a cumulative training volume of 524B tokens for OpenWebText and 65B tokens for LM1B. This fine-tuning required approximately 3 days on 8×A100-80 GB GPUs for OpenWebText and about 36 hours on 2×A100-80 GB GPUs for LM1B.

## B.2 Loss Function

Our training objective consists of two primary loss components: (i) a standard cross-entropy loss computed on the *settled* tokens, and (ii) a diffusion-weighted loss calculated on the *active* tokens. We propose the *Hybrid Diffusion Cross-Entropy* (HDCE) as our diffusion-weighted loss, which blends a per-token cross-entropy loss with a specialized weighting strategy contingent upon whether tokens are masked, shuffled, or unchanged.

Formally, the HDCE loss is defined as:

$$\mathcal{L}_{\text{HDCE}}(\theta) = \frac{1}{Nd} \sum_{i=1}^{N} \sum_{t=1}^{d} w_{i,t} \left[ -\log p_\theta \left( y_{i,t} \mid x_{i,t} \right) \right], \tag{8}$$

where $N$ denotes the batch size, $d$ is the sequence length, and the per-token loss corresponds to the conventional cross-entropy formulation. The token-specific weights $w_{i,t}$ are defined as:

$$w_{i,t} = \begin{cases} \dfrac{1}{p_{\mathrm{mask}}(i,t)} & \text{if } x_{i,t} \text{ is masked,} \\[2mm] \dfrac{\lambda(1-\epsilon)}{1-p_{\mathrm{mask}}(i,t)} & \text{if } x_{i,t} \text{ is unmasked and shuffled,} \\[2mm] \dfrac{\lambda\epsilon}{1-p_{\mathrm{mask}}(i,t)} & \text{if } x_{i,t} \text{ is unmasked and not shuffled,} \end{cases} \tag{9}$$

where $p_{\mathrm{mask}}(i,t)$ is the masking probability for token $x_{i,t}$, and $\lambda$ and $\epsilon$ represent hyperparameters controlling the relative importance of shuffled versus unshuffled tokens.

In practice, we distinguish two model variants based on the employed diffusion-weighted loss. For $\gamma$-variant hybrid models, we adopt the diffusion-weighted denoising score entropy (DWDSE) loss proposed by Lou et al. (2024), denoted as $\mathcal{L}_{\mathrm{DWDSE}}$. Conversely, for our $\epsilon$-variant hybrid models, we use the proposed HDCE loss as defined in Equation 8.

Letting $\mathcal{L}_{\mathrm{CE}}$ represent the cross-entropy loss computed over $\hat{\mathbf{x}}_t^{\mathrm{settled}}$, our overall loss function is thus expressed as:

$$\mathcal{L} = \beta_1\,\mathcal{L}_{\mathrm{CE}}\big(\hat{\mathbf{x}}_t^{\mathrm{settled}}\big) + \begin{cases} \beta_2\,\mathcal{L}_{\mathrm{DWDSE}}\big(\hat{\mathbf{x}}_t^{\mathrm{active}}\big), & \text{for } \gamma\text{-Hybrid,} \\[2mm] \beta_2\,\mathcal{L}_{\mathrm{HDCE}}\big(\hat{\mathbf{x}}_t^{\mathrm{active}}\big), & \text{for } \epsilon\text{-Hybrid,} \end{cases} \tag{10}$$

where $\beta_1, \beta_2 \in \mathbb{R}$ are hyperparameters balancing these two components.

Additionally, since early positions in the sequence tend to become *settled* sooner, we apply a reweighting strategy to normalize the contribution of *settled* tokens at different positions. Specifically, we partition the sequence of length $d$ into blocks of width $\omega$, assigning each token at position $i$ a weight:

$$w(i) = \frac{\lfloor i/\omega \rfloor}{\lceil d/\omega \rceil - 1}, \quad i = 0, 1, \ldots, d-1, \tag{11}$$

with the convention that if $\lceil d/\omega \rceil = 1$, then $w(i) = 1$ for all $i$.

## B.3 Efficient Training and Inference

As mentioned in Sec. 3.3, we take particular care in crafting our attention matrices to enable KV-caching at inference time. These scheme are particularly beneficent when $\omega \ll d$, but naively using an attention matrix such as Fig. 5 would train the diffusion head on only $\omega$ positions while demanding to process on average $d/2$ context tokens. Here we present how we may train the diffusion head on about approximately half the positions, increasing the training-time efficiency by a factor $d/\omega$. Note that, under these efficient schemes, the reweighing of active tokens as given in Eq. (11) is no longer required.

Figures 6–9 provides examples of attention masks that are compatible with the KV-caching scheme presented in Sec. 3.3, while dedicating about half the positions to the denoising task. Light red/blue squares represent positions that are settled, whereas dark red/blue represent positions that are active. In all cases, the top-left part of the matrix has an autoregressive structure, and the production of dark blue positions attends densely on the corresponding dark red inputs as well as the light red inputs that precede them. All cases presume $d = 12$ and $\omega = 4$.

The ALIGNED cases are easier to understand. For $\tau_{\mathrm{Slide}}$, Fig. 6 presents a situation where it was randomly determined that the denoising will be performed on the intervals $j \leq i < \min(j + \omega, d)$ for $j \in \{2, 5, 11\}$. For $\tau_{\mathrm{Block}}$, these starting points $j$ are always multiples of $\omega$, here $j \in \{0, \omega, 2\omega\}$. The light green blocks indicate entries that are not actually involved in the denoising and could thus potentially be eschewed.

Figures 8 and 9 present the corresponding matrices for the SHIFTED configuration. Notice how settled tokens (light red or the gray $\hat{x}^{-1}$) are repeated as the first input of an interval to denoise in the second half of the matrix, and how the last output of each such interval is discarded.

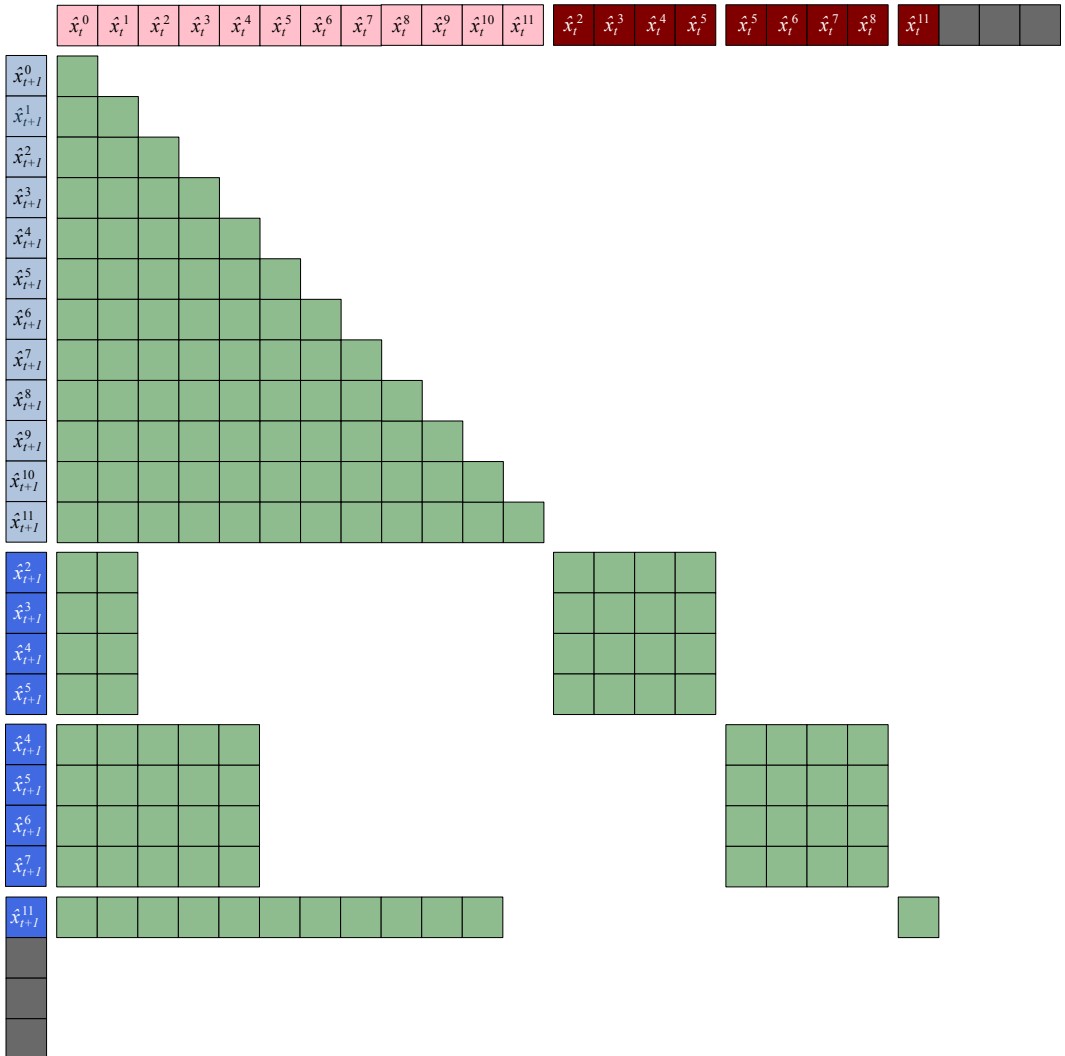

Figure 6: Example training attention mask for ALIGNED configuration for use with $\tau_{\text{Slide}}^{\omega=4}$.

## B.4 Inference and KV-Caching

At inference, both Euler and $\tau$-leaping analytical solutions are available; however, our empirical results suggest that $\tau$-leaping is the de facto superior choice. Because each step "settles" a prefix of tokens, we can cache the corresponding key/value pairs and avoid recomputing them on subsequent passes. Concretely, let

$$L = \text{sequence length}, \quad \rho = \text{tokens generated per step}, \quad \omega = \text{window width}.$$

Then the number of transformer calls is

$$N = \left\lceil (L - \omega)/\rho \right\rceil + 1,$$

and the total token-processing cost is

$$\text{Without cache:} \quad N \times \omega,$$
$$\text{With KV-cache:} \quad \omega + (N - 1)\rho.$$

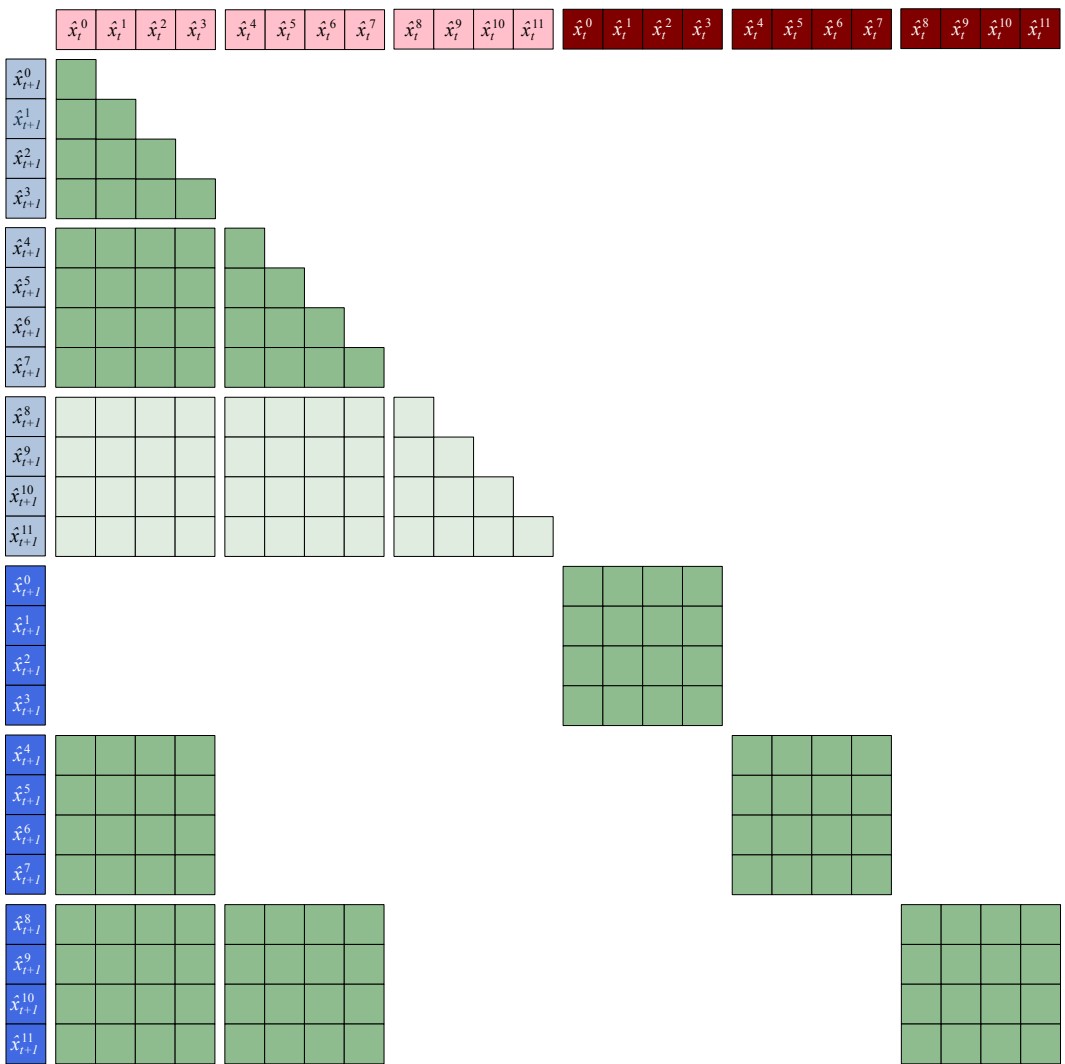

Figure 7: Example training attention mask for ALIGNED configuration for use with $\tau_{\text{Block}}^{\omega=4}$.

## B.5 Weighted token-embedding

Ou et al. (2024) demonstrated that when employing the absorbing transition matrix $Q_{\text{Absorb}}$, scaling the model's score by the analytic, time-dependent factor

$$\frac{\exp(-\bar{\sigma}(t))}{1 - \exp(-\bar{\sigma}(t))}$$

causes the remaining score to be independent of $t$, eliminating the need to explicitly condition on time within the network. However, when using the $\gamma$-Hybrid process $(1 - \gamma)Q_{\text{Absorb}} + \gamma\,Q_{\text{Uniform}}$, this factor remains present but is insufficient for capturing all temporal dependencies. In particular, under the hybrid process, an unmasked token is perturbed with probability $e^{-\gamma\bar{\sigma}(t)}$ (whereas under $Q_{\text{Absorb}}$ alone, a non-mask token remains unchanged).

In other words, when the model encounters an unmasked input token $x_t^i$ subject to cumulative noise $\bar{\sigma}(t)$, it should treat that token as if it were unperturbed with probability $e^{-\gamma\bar{\sigma}(t)}$, and as if it were masked with probability $1 - e^{-\gamma\bar{\sigma}(t)}$. One natural way to embed this inductive bias into the model is to interpolate the token's embedding accordingly. Denoting

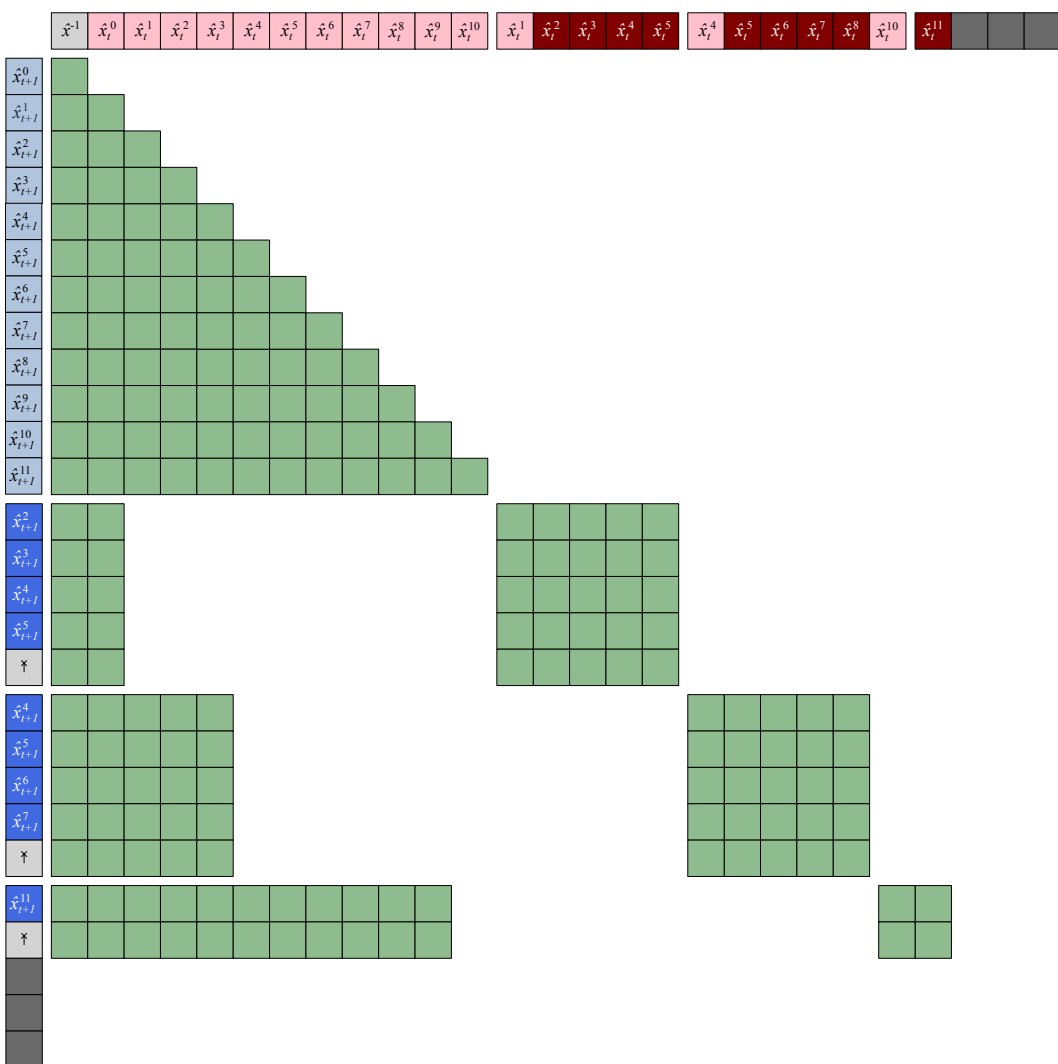

Figure 8: Example training attention mask for SHIFTED configuration for use with $\tau_{\text{Slide}}^{\omega=4}$.

by $\mathbf{f}(x_t^i) \in \mathbb{R}^{d_{\text{model}}}$ the standard embedding of token $x_t^i$, we replace it with

$$e^{-\gamma\bar{\sigma}(t)} \, \mathbf{f}(x_t^i) \, + \, \left(1 - e^{-\gamma\bar{\sigma}(t)}\right) \mathbf{f}(\text{MASK}) \,. \tag{12}$$

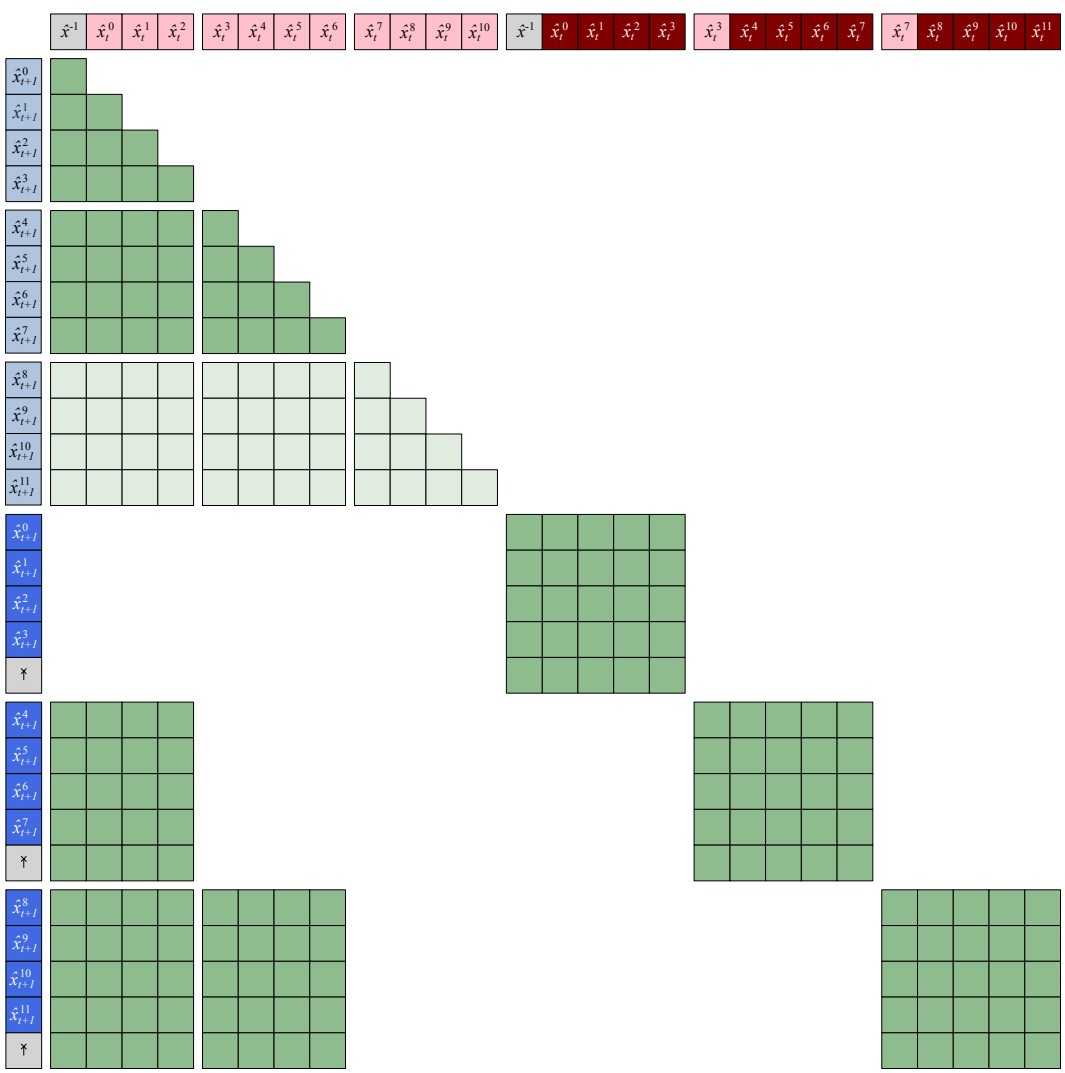

Figure 9: Example training attention mask for SHIFTED configuration for use with $\tau_{\text{Block}}^{\omega=4}$.

## C   Model Evaluation Metrics

### C.1   Upper Bound Estimation of Perplexity

(To simplify the notation, the following uses $\mathbf{y}$ to represent non-noised samples.)

Evaluating perplexity for diffusion-based language models is challenging because the model's likelihood involves an integration over a continuum of noise levels. In our work, we estimate the negative log-likelihood (NLL) via a Monte Carlo (MC) approximation over a discrete set of diffusion timesteps. In particular, given a trained model $p_\theta(\mathbf{y})$ and a diffusion process that perturbs a sequence $\mathbf{y}$ into latent states $\hat{\mathbf{x}}_t$ (with $t \in [0, T]$), our goal is to estimate the per-token loss that, when exponentiated, yields an *upper bound* on the true perplexity.

Let

$$\log p_\theta(\mathbf{y}) = \mathbb{E}_{t \sim q(t)}\Big[\log p_\theta(\mathbf{y} \mid \hat{\mathbf{x}}_t)\Big] - D_{\mathrm{KL}}\Big(q_t(\mathbf{x}_t \mid \mathbf{y}) \,\|\, p(\mathbf{x}_t)\Big).$$

In practice, we approximate the expectation with $M$ Monte Carlo samples:

$$\hat{L}(\mathbf{y}) = \frac{1}{M} \sum_{i=1}^{M} \log p_\theta(\mathbf{y} \mid \hat{\mathbf{x}}_{t_i}), \quad t_i \sim \mathrm{Uniform}(0, 1).$$

Since our loss function returns the *total* NLL over a sequence of length $d$ (i.e., it produces a tensor of per-token losses whose sum over tokens yields the total loss for a sequence), we define the average per-token loss as

$$\ell = \frac{\hat{L}(\mathbf{y})}{d}.$$

The estimated perplexity is then given by

$$\mathrm{PPL} = \exp(\ell).$$

Because the MC approximation truncates the integration to $M$ samples, the resulting perplexity is an upper bound on the true perplexity. In our experiments, we typically set $M = 1000$, and we observe that increasing $M$ further reduces the variance of the estimate.

For generators with semi-autoregressive or autoregressive configurations (with a fixed window width $\omega$), we calculate the NLL only over the $\tau_t^{\mathrm{active}}$ tokens (i.e., those tokens that are actively being updated during generation). This ensures that the perplexity computation is fair and reflects the model's performance on the tokens whose values are uncertain, rather than being diluted by tokens that are already settled.

We summarize the estimation procedure in Algorithm 1.

**Remarks 1.** *In our setting, the loss function $\mathcal{L}(\theta, \mathbf{y}, t)$ returns a tensor of shape $[B, d]$. If the generator is semi-autoregressive with a fixed active window of width $\omega$, then the loss is computed only on the $\tau_t^{active}$ tokens. The expectation over t is approximated by averaging over M independent samples. Since the integral is truncated, the computed perplexity serves as an* upper bound *on the true value. Finally, dividing the total loss by $N_{total}$ gives the average per-token loss, so that the perplexity is computed as* $\exp(avg\ loss\ per\ token)$.

**Remarks 2.** *All our "Test PPL" metrics are computed using the loss presented above, i.e., **not** the one presented in Appendix B.2. For this reason, the upper bound provided by Algorithm 1 assesses the quality of the neural network at language modeling. In particular, it does not account for the full inference strategy. Therefore, although the Test PPL metric may **indirectly** be affected by the training objective (described in Appendix B.2), the choice of hyperschedule and the values of $\epsilon$ or $\gamma$ through their combined impact on the training of the neural network's weights, these hyperparameters are **not** directly involved when running Algorithm 1.*

---

**Algorithm 1** Monte Carlo Upper Bound Estimation of Perplexity

---

**Require:** Model parameters $\theta$, loss function $\mathcal{L}$, evaluation dataset $\mathcal{D}$, number of MC samples $M$, sequence length $d$
1: Initialize total loss $\mathcal{L}_{\text{total}} \leftarrow 0$ and token count $N_{\text{total}} \leftarrow 0$
2: **for** each batch $\mathbf{y} \in \mathcal{D}$ **do**
3:     Initialize batch loss $\mathcal{L}_{\text{batch}} \leftarrow 0$
4:     **for** $i = 1$ to $M$ **do**
5:         Sample $t_i \sim \text{Uniform}(0, 1)$
6:         Compute per-token loss $\ell_i \leftarrow \mathcal{L}(\theta, \mathbf{y}, t_i) \in \mathbb{R}^{B \times d}$
7:         $\mathcal{L}_{\text{batch}} \leftarrow \mathcal{L}_{\text{batch}} + \ell_i$
8:     **end for**
9:     $\mathcal{L}_{\text{batch}} \leftarrow \frac{1}{M} \mathcal{L}_{\text{batch}}$                    ▷ Average over MC samples
10:     $\mathcal{L}_{\text{total}} \leftarrow \mathcal{L}_{\text{total}} + \sum \mathcal{L}_{\text{batch}}$
11:     $N_{\text{total}} \leftarrow N_{\text{total}} + B \times d$
12: **end for**
13: Compute average per-token loss: $\ell = \mathcal{L}_{\text{total}} / N_{\text{total}}$
14: Compute perplexity: $\text{PPL} \leftarrow \exp(\ell)$ **return** PPL

---

## C.2   Generative Perplexity Evaluation

In addition to the intrinsic perplexity estimation described in Section C.1, we also assess our generator via *generative perplexity*. In this approach, a pretrained autoregressive language model — in our case, GPT2-Large Radford et al. (2019) — serves as an external judge of the generated sequences. This method has been used in prior work Keskar et al. (2019); Holtzman et al. (2020) as a proxy for fluency and coherence when direct likelihood evaluation is intractable.

Concretely, our procedure is as follows: We first sample sequences from our diffusion generator using the analytical solution Lou et al. (2024). Since our generator and GPT2-Large may employ different tokenization schemes, the generated samples are retokenized using the GPT2 tokenizer. The retokenized sequences are then fed into the GPT2-Large model, which computes the negative log-likelihood (NLL) for each token; this value quantifies the *surprise* of the judge regarding the generated text. Finally, by averaging the NLL over all tokens and exponentiating the result, we obtain the generative perplexity:

$$\text{PPL}_{\text{gen}} = \exp\left( \frac{1}{N} \sum_{i=1}^{N} -\log p_{\text{GPT2}}(x_i) \right),$$

where $N$ is the total number of tokens in the generated text and $p_{\text{GPT2}}(x_i)$ denotes the probability assigned by GPT2-Large to token $x_i$.

In the case of semi-autoregressive or autoregressive models with a fixed active window of width $\omega$, we compute the NLL only over the tokens corresponding to the active portion $\tau_t^{\text{active}}$. This ensures that the perplexity is estimated fairly by focusing on those positions where the model is actually making nontrivial predictions.

The full procedure is summarized in Algorithm 2.

---

**Algorithm 2** Generative Perplexity Estimation via External Judge

---

**Require:** Generator $G$ with parameters $\theta$, pretrained judge model $J$ (GPT2-Large), number of samples $S$

1: Generate a set of samples $\{\mathbf{y}^{(s)}\}_{s=1}^{S}$ using the analytical solution of $G$
2: Retokenize each generated sample using the GPT2 tokenizer:

$$\tilde{\mathbf{y}}^{(s)} \leftarrow \text{Tokenize}_{\text{GPT2}}(\mathbf{y}^{(s)})$$

3: **for** each retokenized sample $\tilde{\mathbf{y}}^{(s)}$ **do**
4:     Compute per-token negative log-likelihood $\ell^{(s)} \leftarrow -\log p_J(\tilde{\mathbf{y}}^{(s)})$
5: **end for**
6: Compute the overall average per-token loss:

$$\bar{\ell} = \frac{1}{N} \sum_{s=1}^{S} \ell^{(s)}$$

7: Compute generative perplexity:

$$\text{PPL}_{\text{gen}} = \exp(\bar{\ell})$$

   **return** $\text{PPL}_{\text{gen}}$

---

# D   Adaptive Correction Sampler (ACS) Pseudo Code

For completeness, we present pseudo code for both the original sampling procedure and our proposed Adaptive Correction Sampler (ACS).

---

**Algorithm 3** Original Sampler

---

1: **Input:** model, context length $L$, total steps $S$, temperature $T$.
2: Initialize $x \leftarrow$ a tensor of shape $[B, L]$ filled with mask tokens.
3: Compute timesteps $\{t_i\}_{i=0}^{S}$ linearly spaced between 1 and 0).
4: **for** $i = 0, \ldots, S - 1$ **do**
5:     Set $t \leftarrow t_i$ and $s \leftarrow t_{i+1}$
6:     Compute transfer probability $p_{\text{transfer}} \leftarrow 1 - \frac{s}{t}$
7:     **for** each token in $x$ **do**
8:         **if** token is masked and a random draw is below $p_{\text{transfer}}$ **then**
9:             Add Gumbel noise to the token's logits and update it via $\arg\max$.
10:         **end if**
11:     **end for**
12:     Update $x$ accordingly.
13: **end for**
14: **return** $x$

---

---

**Algorithm 4** Adaptive Correction Sampler (ACS)

---

1: **Input:** model, context length $L$, total steps $S$, temperature $T$, correction parameter $\eta$.
2: Initialize $x \leftarrow$ a tensor of shape $[B, L]$ filled with mask tokens.
3: Compute timesteps $\{t_i\}_{i=0}^{S}$ linearly spaced between 1 and 0.
4: **for** $i = 0, \ldots, S-1$ **do**
5:     Set $t \leftarrow t_i$ and $s \leftarrow t_{i+1}$
6:     Compute transfer probability $p_{\text{transfer}} \leftarrow 1 - \frac{s}{t}$
7:     **for** each token in $x$ **do**
8:         **if** token is masked and a random draw is below $p_{\text{transfer}}$ **then**
9:             Update the token using the standard denoising update (with Gumbel noise).
10:         **else if** token is unmasked and a random draw is below $\eta\,(1 - p_{\text{transfer}})$ **then**
11:             Update the token via a uniform correction mechanism.
12:         **end if**
13:     **end for**
14:     Update $x$ accordingly.
15: **end for**
16: **return** $x$

---

# E    Experimental Setup

In our experiments, we adopt a training and evaluation protocol similar to that of Sahoo et al. Sahoo et al. (2024). We conduct experiments on two datasets: the One Billion Word Benchmark (*LM1B*; Chelba et al. (2014)) and *OpenWebText* (*OWT*; Gokaslan et al. (2019)). For models trained on LM1B, we employ the `bert-base-uncased` tokenizer with a context length of $d = 128$ tokens, and report perplexities on the test split of *LM1B*. In contrast, models trained on *OWT* use the `GPT2` tokenizer Radford et al. (2019) with a context length of $d = 1024$ tokens.

Since the *LM1B* corpus predominantly consists of single-sentence examples, a straightforward padding scheme for reaching a fixed context length may not be optimal. Accordingly, following Sahoo et al. Sahoo et al. (2024), we concatenate and wrap sequences to fit a context window of 128 tokens. Similarly, for *OWT* we concatenate and wrap sequences to 1024 tokens, rather than simply truncating or padding, thereby ensuring that our evaluation is performed on coherent text segments. In the case of *OWT*, which lacks a designated validation split, we reserve the final 100K documents for validation purposes.

Our model architecture builds upon the diffusion transformer framework Peebles & Xie (2023), augmented with rotary positional embeddings Su et al. (2024). We instantiate our autoregressive baselines — SEDD, MDLM — with a transformer backbone as described in Sahoo et al. (2024): 12 layers, a hidden dimension of 768, and 128 attention heads.

# F    Additional Results

This section presents additional results and ablation studies on our family of models.

## F.1    Zeroshot Perplexity

In Table 5 we report the full list of results for the family of our models against all the reported models available online. This is an extensive and more complete version of Table 3.

## F.2    Generative Perplexity Using a Judge LLM

In this subsection, we report the *generative perplexity* (see Section C.2) of our model. Zheng et al. (2024) were the first to demonstrate that the generative perplexity evaluations of baseline masked diffusion language models are flawed due to imprecise categorical sampling. They show that employing 32-bit floating point precision in categorical sampling via the Gumbel trick induces an artificial temperature-lowering effect, which results in a lower (i.e.,

| Method | PTB | WikiText | LM1B | Lambada | AG News | Pubmed | Arxiv |
|---|---|---|---|---|---|---|---|
| GPT-2 (WebText)[*] | 138.43 | 41.60 | 75.20 | 45.04 | – | – | – |
| Transformer (Sahoo et al., 2024) | 82.05 | 25.75 | 51.25 | 51.28 | 52.09 | 49.01 | 41.73 |
| D3PM[†] (Austin et al., 2021) | 200.82 | 75.16 | 138.92 | 93.47 | – | – | – |
| Plaid[†] (Gulrajani & Hashimoto, 2024) | 142.60 | 50.86 | 91.12 | 57.28 | – | – | – |
| MD4 (Shi et al., 2024) | 102.26 | 35.90 | 68.10 | 48.43 | – | – | – |
| SEDD Absorb[‡] (Lou et al., 2024) | 96.33 | 35.98 | 68.14 | 48.93 | 67.82 | 45.39 | 40.03 |
| MDLM[‡] (Sahoo et al., 2024) | 90.96 | 33.22 | 64.94 | 48.29 | 62.78 | 43.13 | 37.89 |
| BD3-LM $L' = 4$ (Arriola et al., 2025) | 96.81 | 31.31 | **60.88** | 50.03 | **61.67** | 42.52 | 39.20 |
| RADD-$\lambda$-DCE (Ou et al., 2024) | 107.85 | 37.98 | 72.99 | 51.70 | – | – | – |
| $\gamma$-Hybrid (444B) [$\gamma= 0.01, \tau_{\text{Flat}}$, ALIGNED] | **89.94** | **30.02** | 61.01 | **45.38** | 67.51 | 46.57 | 40.62 |
| $\epsilon$-Hybrid (444B) [$\epsilon= 0.01, \tau_{\text{Flat}}$, ALIGNED] | **90.89** | 32.53 | 68.91 | 50.23 | 64.61 | **41.18** | **37.85** |
| $\gamma$-Hybrid (444B) [$\gamma= 0.01, \tau_{\text{Flat}}$, SHIFTED] | 100.88 | 37.48 | 71.51 | 56.57 | 70.69 | 43.06 | 38.83 |
| $\gamma$-Hybrid [$\gamma= 0.01, \tau_{\text{Slide}}^{\omega=d/4}$, ALIGNED] | **90.67** | 31.73 | 73.71 | **50.03** | 68.27 | **41.49** | **37.89** |
| $\gamma$-Hybrid [$\gamma= 0.01, \tau_{\text{Block}}^{\omega=d/4}$, ALIGNED] | **95.32** | 38.94 | 70.49 | **48.18** | 67.32 | 44.23 | 42.78 |
| $\gamma$-Hybrid [$\gamma= 0.01, \tau_{\text{Block}}^{\omega=d/64}$, ALIGNED] | **90.74** | 35.24 | 62.64 | 51.21 | 69.62 | **41.46** | **37.13** |
| $\gamma$-Hybrid [$\gamma= 0.01, \tau_{\text{Block}}^{\omega=d/64}$, SHIFTED] | 95.22 | 32.64 | 63.68 | **44.75** | 62.18 | **42.01** | 37.33 |

Table 5: Zero-shot unconditional perplexity on seven benchmark datasets from Lou et al. (2024) and Sahoo et al. (2024) and Arriola et al. (2025). [†]Reported in He et al. (2022). [‡]Reported in Arriola et al. (2025). [*]The GPT-2 numbers are reported for the checkpoint pretrained on WebText and are not a direct comparison. All models are trained for 524B tokens unless otherwise stated. All diffusion models are upper bounds; the best diffusion value is **bolded**, the second best values is underscored.

seemingly better) generative perplexity at the expense of reduced entropy—a key indicator of generation diversity. Their proposed remedy is to cast the values to 64-bit floating point precision.

To ensure a fair comparison of generative perplexity across baselines, we report both the flawed (32-bit) and the corrected (64-bit) perplexity values in Table 6. All the entries were resampled using full precision. Our results indicate that, irrespective of the artificial temperature effect, our models consistently outperform all diffusion-based counterparts.

| Method | FP32-PPL $\downarrow$ | FP64-PPL $\downarrow$ |
|---|---|---|
| SEDD-Absorb [14] | 43.41 | 105.91 |
| MDLM [21] | 43.88 | 108.88 |
| $\gamma$-Hybrid [$\gamma= 0.05, \tau_{\text{Flat}}$, ALIGNED] (444B tokens) | **39.53** | **89.05** |
| $\gamma$-Hybrid [$\gamma= 0.01, \tau_{\text{Flat}}$, SHIFTED] (444B tokens) | 48.08 | 110.60 |
| $\gamma$-Hybrid [$\gamma= 0.01, \tau_{\text{Block}}^{\omega=d/4}$, ALIGNED] | **40.48** | **85.01** |
| $\gamma$-Hybrid [$\gamma= 0.01, \tau_{\text{Slide}}^{\omega=d/4}$, ALIGNED] | 61.05 | 131.45 |
| $\gamma$-Hybrid [$\gamma= 0.01, \tau_{\text{Block}}^{\omega=d/64}$, ALIGNED] | 76.12 | 121.99 |
| $\gamma$-Hybrid [$\gamma= 0.01, \tau_{\text{Block}}^{\omega=d/4}$, SHIFTED] | 53.89 | 111.73 |

Table 6: Generative perplexities (PPL; lower is better) on OWT. All models were trained for 524B tokens unless otherwise indicated. "FP32" denotes the flawed 32-bit sampling, whereas "FP64" corresponds to the corrected 64-bit precision values. All available models were resampled using their published weights.

## F.3 Inference Pareto frontier Results

In Table. 7 we report the results we utilized to report the Pareto frontier plots in the main paper.

| **Higher $\rho$ values ($\rho = 8, 4$)** | | | | | | |
|---|---|---|---|---|---|---|
| Method | MAUVE ($\uparrow$) | | Gen PPL. ($\downarrow$) | | Entropy ($\uparrow$) | |
| | $\rho = 8$ | $\rho = 4$ | $\rho = 8$ | $\rho = 4$ | $\rho = 8$ | $\rho = 4$ |
| SEDD | 0.410 | 0.491 | 139.2 | 130.1 | 5.72 | 5.63 |
| MDLM | 0.921 | 0.959 | 128.5 | 116.4 | 5.63 | 5.58 |
| $\gamma$-Hybrid [$\gamma = 0.05, \tau_{\text{Flat}}$, ALIGNED] | 0.809 | 0.817 | 85.9 | 89.5 | 5.37 | 5.38 |
| $\gamma$-Hybrid [$\gamma = 0.01, \tau_{\text{Flat}}$, SHIFTED] | 0.666 | 0.700 | 99.2 | 93.9 | 5.46 | 5.45 |
| $\gamma$-Hybrid [$\gamma = 0.01, \tau_{\text{Slide}}^{\omega=d/4}$, ALIGNED] | 0.775 | 0.788 | 107.3 | 106.0 | 5.53 | 5.53 |
| $\epsilon$-Hybrid [$\epsilon = 0.01, \tau_{\text{Flat}}$, ALIGNED] | 0.848 | 0.928 | 84.2 | 69.8 | 5.36 | 5.33 |
| $\epsilon$-Hybrid [$\epsilon = 0.01, \tau_{\text{Block}}^{\omega=d/4}$, ALIGNED] | 0.964 | 0.811 | 104.2 | 76.9 | 5.42 | 5.25 |
| **Lower $\rho$ values ($\rho = 2, 1$)** | | | | | | |
| Method | MAUVE ($\uparrow$) | | Gen PPL. ($\downarrow$) | | Entropy ($\uparrow$) | |
| | $\rho = 2$ | $\rho = 1$ | $\rho = 2$ | $\rho = 1$ | $\rho = 2$ | $\rho = 1$ |
| SEDD | 0.512 | 0.457 | 127.2 | 126.8 | 5.60 | 5.58 |
| MDLM | 0.947 | 0.897 | 115.8 | 108.8 | 5.61 | 5.60 |
| $\gamma$-Hybrid [$\gamma = 0.05, \tau_{\text{Flat}}$, ALIGNED] | 0.877 | 0.895 | 97.9 | 96.8 | 5.40 | 5.41 |
| $\gamma$-Hybrid [$\gamma = 0.01, \tau_{\text{Flat}}$, SHIFTED] | 0.728 | 0.744 | 96.4 | 93.9 | 5.45 | 5.47 |
| $\gamma$-Hybrid [$\gamma = 0.01, \tau_{\text{Slide}}^{\omega=d/4}$, ALIGNED] | 0.553 | 0.819 | 105.5 | 100.2 | 5.46 | 5.41 |
| $\epsilon$-Hybrid [$\epsilon = 0.01, \tau_{\text{Flat}}$, ALIGNED] | 0.957 | 0.947 | 61.3 | 43.9 | 5.28 | 5.18 |
| $\epsilon$-Hybrid [$\epsilon = 0.01, \tau_{\text{Block}}^{\omega=d/4}$, ALIGNED] | 0.813 | 0.916 | 71.7 | 59.1 | 5.38 | 5.25 |

Table 7: Sample quality of absorbing state discrete diffusion models. Upper block: higher $\rho$ values ($\rho = 8, 4$); Lower block: lower $\rho$ values ($\rho = 2, 1$).

## F.4 Effect of Varying $\gamma$

In this section, we examine the influence of the hyperparameter $\alpha$, which modulates the contribution of $Q_{\text{uniform}}$ in the hybrid process and thereby allows the model to reexamine its predictions after unmasking a token. As shown in Table 8, while the corrective influence of $Q_{\text{uniform}}$ is essential, the value of $\alpha$ must remain relatively small. If $\alpha$ is set too high, the model tends to simply reshuffle the tokens and the MASK token, effectively undermining the intended unmasking process.

Another perspective is that increasing $\alpha$ reduces the penalty associated with errors in the unmasking operation, thereby devaluing its corrective impact. Moreover, during the denoising process, each token is influenced not only by its own prediction but also by

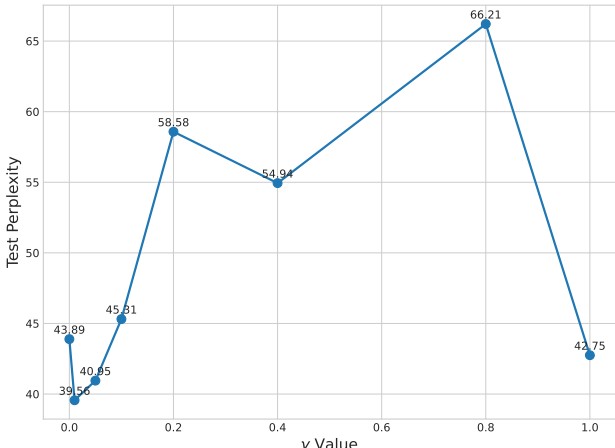

Figure 10: Ablation study illustrating the effect of varying the $\gamma$ (for $\gamma$-Hybrid variants) parameter on perplexity evaluated on the OWT test split at the 26B-token observation point. Lower perplexity values reflect improved model performance. Consistent with our previous observations, $\gamma$ between 0.01 and 0.1 yields optimal performance.

the context provided by neighboring tokens. Consequently, if a token (e.g., token $A$) is mispredicted, the resulting change in the overall structure may leave little opportunity for subsequent correction.

**Remarks 3.** *The analyses presented above are mainly intuitive. Further empirical investigation is necessary to confirm.*

| Configuration | FP64-PPL $\downarrow$ |
|---|---|
| $\gamma= 0.01$ | 84.32 |
| $\gamma= 0.05$ | 82.27 |
| $\gamma= 0.1$ | 85.15 |
| $\gamma= 0.4$ | 91.48 |
| $\gamma= 0.8$ | 90.99 |

Table 8: Generative perplexities (PPL; lower is better) on OWT. All the models are trained under ALIGNED configuration with $\tau_{\text{Block}}^{\omega=256}$ for 524B tokens. We use the double precision, denoted as "FP64-PPL".

To further examine the effect of $\alpha$, we evaluate our trained models' test perplexity (on OWT held out set) after processing 26B tokens under various configurations. As shown in Fig. 10, small $\alpha$ values—approximately 0.01 and 0.1—yield the best performance.

### F.5   Effect of Varying $\rho$

In this section, we examine the impact of varying $\rho$ in our $\gamma$-Hybrid models using the $\tau_{\text{Block}}^{\omega=256}$ hyperschedule. Recall that the parameter $\rho$ is modified during the generation process and directly only influences the quality of the generated sequence. Consequently, we adopt *generative perplexity* as our evaluation metric. For completeness—and to facilitate comparison with prior baselines—we report the generative perplexity computed under both double precision (FP64) and full precision (FP32), as illustrated in Table 9. As $\rho$ decreases, the generation process becomes slower, thereby entering the "think hard" regime; in this regime, the model tends to produce higher-quality outputs at the cost of increased computational time.

**Remarks 4.** *This trade-off is a key characteristic of diffusion models, which inherently possess a flexible inductive bias that allows for varying degrees of commitment in generation. In contrast, autoregressive models are restricted to generating one token at a time.*
*Moreover, under the flawed 32-bit sampling scheme, increasing the number of sampling steps effectively reduces the artificial temperature, thereby reducing the tokenwise entropy. In contrast, the tokenwise entropy remains mostly unaffected when the Gumbel trick is executed in double precision.*

| Configuration | FP32-PPL $\downarrow$ | FP64-PPL $\downarrow$ |
|---|---|---|
| $\rho = 16\ (T = 64)$ | 75.57 | 100.11 |
| $\rho = 8\ (T = 128)$ | 64.90 | 95.38 |
| $\rho = 4\ (T = 256)$ | 53.02 | 81.37 |
| $\rho = 2\ (T = 512)$ | 44.15 | 79.3 |
| $\rho = 1\ (T = 1024)$ | 40.48 | 85.01 |
| $\rho = \frac{1}{2}\ (T = 2048)$ | 33.09 | 87.31 |
| $\rho = \frac{1}{4}\ (T = 4096)$ | 25.39 | 88.75 |
| $\rho = \frac{1}{8}\ (T = 8192)$ | 24.05 | 83.21 |

Table 9: Generative perplexities (PPL; lower is better) on OWT. The same very model ($\gamma$-Hybrid [$\gamma = 0.01$, $\tau_{\text{Block}}^{\omega=d/4}$, ALIGNED]) has been used under different generation regimes. $\rho$ value as well as equivalent $T$ "diffusion steps" are used in the table. "FP32" denotes the flawed 32-bit sampling, whereas "FP64" corresponds to the corrected 64-bit precision values.

### F.6   Effect of Varying $\eta$ in Adaptive Correction Sampler

In this section, we investigate the effect of the hyperparameter $\eta$ in our proposed Adaptive Correction Sampler. Table 10 illustrates results for our $\epsilon$-variety family of models. As we increase the number of sampling steps (corresponding to a decrease in $\rho$), our model tends to overcorrect when $\eta$ is too large, which ultimately harms generation diversity—marked as "extreme values" in the table.. To mitigate this issue, we find that using smaller $\eta$ values is beneficial. We further suggest that the optimal choice of $\eta$ is related to the $\epsilon$ value used during training: the more inherently corrective the model is, the smaller the optimal $\eta$ should be.

### F.7   Inference Speed Up with Caching

In Table 11, we report the wallclock time required to generate eight samples for various models on a single *NVIDIA A100 80 GB* GPU. Owing to our custom attention masks, we were unable to leverage fast transformer kernels such as Flash-Attention, which in turn results in slower sampling speeds. We note that future work may design specialized kernels that are compatible with our attention masks to accelerate inference.

| Model Family | Sampler | MAUVE (↑) | | | | Gen PPL. (↓) | | | | Entropy (↑) | | | |
|---|---|---|---|---|---|---|---|---|---|---|---|---|---|
| | | $\rho=8$ | $\rho=4$ | $\rho=2$ | $\rho=1$ | $\rho=8$ | $\rho=4$ | $\rho=2$ | $\rho=1$ | $\rho=8$ | $\rho=4$ | $\rho=2$ | $\rho=1$ |
| $\epsilon$-Hybrid [$\epsilon=0.01$, $\tau_{\text{Flat}}$, ALIGNED] | Original Sampler | 0.950 | 0.944 | 0.848 | 0.779 | 130.78 | 124.75 | 121.90 | 129.52 | **5.51** | 5.47 | **5.49** | **5.50** |
| | ACS ($\eta=0.25$) | 0.955 | 0.821 | 0.859 | 0.928 | **79.64** | 65.06 | **55.05** | 49.09 | 5.35 | 5.28 | 5.24 | 5.19 |
| | ACS ($\eta=0.05$) | 0.846 | 0.99 | 0.865 | 0.936 | 105.94 | 93.91 | 83.83 | 77.16 | 5.46 | **5.48** | 5.31 | 5.29 |
| | ACS ($\eta=0.01$) | 0.848 | 0.928 | 0.957 | 0.947 | 84.28 | 69.84 | 61.35 | **43.98** | 5.36 | 5.33 | 5.28 | 5.18 |
| | ACS ($\eta=0.001$) | 0.871 | 0.949 | **0.919** | **0.998** | 80.37 | **64.42** | 55.48 | 45.80 | 5.35 | 5.31 | 5.25 | 5.15 |
| | extreme values | | | | | | | | | | | | |
| | ACS ($\eta=0.99$) | 0.618 | 0.582 | 0.754 | 0.508 | 28.05 | 24.94 | 22.01 | 21.26 | 4.98 | 4.86 | 4.66 | 4.41 |
| | ACS ($\eta=0.75$) | 0.883 | 0.770 | 0.999 | 0.293 | 28.51 | 24.57 | 23.49 | 22.23 | 5.03 | 4.90 | 4.79 | 4.56 |
| | ACS ($\eta=0.5$) | 0.740 | 0.987 | 0.859 | 0.837 | 31.86 | 64.42 | 55.48 | 45.80 | 5.35 | 5.31 | 5.25 | 5.21 |
| $\epsilon$-Hybrid [$\epsilon=0.01$, $\tau_{\text{Block}}^{\omega=d/4}$, ALIGNED] | Original Sampler | 0.916 | **0.976** | 0.778 | 0.847 | 148.45 | 130.16 | 139.64 | 142.13 | 5.39 | 5.35 | 5.43 | 5.46 |
| | ACS ($\eta=0.25$) | 0.962 | 0.948 | 0.652 | 0.653 | 112.87 | **64.01** | **54.67** | **43.32** | 5.34 | 5.13 | 5.01 | 4.78 |
| | ACS ($\eta=0.05$) | 0.964 | 0.811 | 0.813 | 0.916 | **104.26** | 76.98 | 71.77 | 59.15 | 5.42 | 5.25 | 5.38 | 5.25 |
| | ACS ($\eta=0.01$) | 0.568 | 0.746 | 0.767 | **0.974** | 144.71 | 107.06 | 101.30 | 75.91 | 5.53 | 5.30 | 5.38 | 5.35 |
| | ACS ($\eta=0.001$) | **0.979** | 0.847 | **0.977** | 0.906 | 150.41 | 150.32 | 139.66 | 114.12 | **5.48** | **5.54** | **5.55** | **5.48** |

Table 10: Sample quality of $\epsilon$-variant models using different samplers.

Our $\tau_{\text{Block}}^{\omega}$ models are intrinsically faster since, during sampling, only the tokens corresponding to the $\tau_{\text{settled}}$ and $\tau_{\text{active}}$ components are fed into the transformer (e.g., $\omega, 2\omega, \ldots$, up to $d$ tokens). Moreover, incorporating KV-caching would further boost the sampling speed.

| | *Time*(↓) |
|---|---|
| SEDD | 68 |
| MDLM | 59 |
| + caching | 36 |
| $\gamma$-Hybrid [$\tau_{\text{Flat}}$, ALIGNED] | 131 |
| $\gamma$-Hybrid [$\tau_{\text{Block}}^{\omega=d/4}$, ALIGNED] | 79 |
| + KV-caching | 38 |

Table 11: Wall clock time reported in seconds to generate 8 samples on a single NVIDIA A100 80 GB GPU. The same number of diffusion steps were utilized for all the models.

## F.8 Training Throughput Evaluation

Table 12 compares training throughput of our $\epsilon$-Hybrid [$\epsilon=0.01$, $\tau_{\text{Block}}^{\omega=d/4}$, ALIGNED] in both non-efficient and efficient settings, alongside MDLM (Sahoo et al., 2024), SEDD (Lou et al., 2024), and an AR baseline, on a single *NVIDIA A100 80 GB* GPU. We use the largest batch size divisible by two (64 for most methods; 32 for the efficient variant due to its "double-length context" strategy) and set the window size to $\omega=d/4$ (smaller $\omega$ further amplifies the training efficiency gains).

By leveraging our efficient attention masks and doubling the context length via concatenation of clean and noisy sequences, the efficient variant achieves nearly twice the active-token throughput of its non-efficient counterpart. This demonstrates that our hyperschedule variants incur substantially lower training cost while matching the effective throughput of other diffusion baselines.

| Method | Throughput (token/ms) | Effective Throughput (active token/ms) |
|---|:---:|:---:|
| $\epsilon$-Hybrid $[\epsilon = 0.01,\ \tau_{\text{Block}}^{\omega=}{}^{d}/4,\ \text{ALIGNED}]$ (Non-Efficient) | 100.31 | 25.07 |
| $\epsilon$-Hybrid $[\epsilon = 0.01,\ \tau_{\text{Block}}^{\omega=}{}^{d}/4,\ \text{ALIGNED}]$ (Efficient) | 88.04 | 44.02 |
| MDLM (Sahoo et al., 2024) | 97.64 | 54.42 |
| SEDD (Lou et al., 2024) | 99.29 | 56.01 |
| AR | 127.65 | 127.65 |

Table 12: Training throughput comparison on a single `NVIDIA A100-80` GB GPU. Effective throughput counts only the active tokens processed by the transformer.

## G  Samples

The following is a random selection of a few samples from our 3 diffusion language model families.

### G.1  Undonditonal Sample

Twenty-something host will soon have a new spot at a main commercial-run drugmakers' clinic in his home state of New Mexico.

Chris Brewster is convinced that it could lead to the world market for Type II's genetically modified metallomics drugs from certain not-too-famous contenders elsewhere in the medical marijuana industry who have partnered up with the cannabis company Little Troy Farms to roll out FDA approval for the widely used CRISPR-free Jacoby-Glynda hybrid product.

At least three manufacturers have been selected for the enterprising position, which also includes Schluge Therapeutics, Porsale, Mosaic, Marin and IE Pharma. 'The total number of drug prices in America in 2013 had the highest in any single year.'

Prasepalan, Dr. Dave Rouzeau's permission to begin selling medicine has long been a crusader for medical marijuana and these days, Dr. Brewster and Dr. Rimelter are taking their criminal flair to the next level.

Rouxau and his fellow New Mexico drug regulatory experts will be making their cases in October, which would complete a 90-page interface presenting the medical world, checks and balances and system reviews for each company's patented drugs involved in overall delivery by patient care providers and regulators.

"We have seen this is a breakthrough that patents have exploded," said Johnson, who represents Prasepalan's chiropractor, Adderall, and his partner, Rick Kelbyck, who specializes in pharmaceutical treatment and drug treatment products at Wellesley and Kroll Laboratories, among other giants, in Austin.

"We do have a case to make," and an additional 12 to 25 miles away from Prasepalan, Dr. Hansen will charge a direct fee to Symantec to make the prescription of Nathur Shemogood's Marin-REX Therapeutics-Lucenti, a New Mexico-based machine that has shipped a billion dollars worth of medicine to every business in the United States and Canada.¡—endoftext—¿No man on the verge of retirement must fear the official automated deportation system, fair trade prosecutor Rafaela Gonzalez thinks.

D.C.—a remote northern island nation governed by rigid immigration rules with 90 million illegal immigrants, 85 million people without adequate food, and 1 million illegal, unauthorized immigrants—that will soon surge around the United States?

Gonzalez got the idea from a journalist at a Caution Center paper to mark the anniversary of the United Nation's signing of the massive Comprehensive Economic and Policy Agreement (COP21) in 1995. Typically it's a minor-level agreement that provides an objective, stable key equation wherein every country, every country can control both the economy and country's external affairs by changing its immigration and customs policy. Democracy and separation of the people and law enforcement.

In March of this year, Gonzalez released a new chapter in her career-spanning memoir Magic of Arrows: The Filibuster of Gang Targets and Promise of Perfection. Modeled after a projection of long-term economic effects from government handouts to its leaflet, her book is dated May 21–24: Four days before America secedes from formally abolishing the six-nation customs union, America's ex-military leaders vow to force America to break off the military response to escalating pro-secularism.

The first chapter is rated as one of the first lists of the U.S. Category A countries for its purposes, code for "unfair advantage," with a stylized listing of 22 whose influence has been ascendant since the dawn of capitalism. As Gonzalez points out, the list displays the position of bodies such as United States Samoa and Croatia, whose remotes are sprinkled with their own market share and have expressed renewed interest in reaching out to countries like the BRICS, a 33-nation grouping that then can take tougher measures against those competitors.

"There are tremendous opportunities," Gonzalez writes. "From a Washington perspective, it's easy to see players–the world's poorest, country's most vulnerable–already feeling the need for a more calculated and conscientious policy in regard to smuggling drugs and other illegal trade into our midst."

D.C. is not alone: The Pan American Conference of Scholars announced that more than 3,000 countries sent almost 1,000 requests for tickets, and more than 17,000 applications for tickets have been denied since March of this year. Last May, the Selena–Ranich Trade Union Confederation—the country's only market, for which agribusiness corporations receive billions of dollars of grants annually—received a "delusion team response" of 6,000 applicants in 81 international training centres.

Robert Dellinger, deputy director for the International Union of Red Cross Office for Latin American Mission who awaits a decision in the file

Figure 11: Unconditional samples generated by $\gamma$-Hybrid [$\gamma = 0.01$, $\boldsymbol{\tau}_{\text{Flat}}$, ALIGNED] trained on **OWT** dataset.

SALT LAKE CITY — The Utah Legislature committee chairwoman says it is a problem to persuade the state to legalize drug behavior.

Wroeff Bugler reports for the Salt Lake Tribune Jan. 7, 2014 The bill's sponsor, Rep. Mike Tate, chairman of the Utah Business Improvement Association, says they would legalize the use of drugs "in a variety of policy areas." A Republican lawmaker called them to say a ban on the use of marijuana is the best way to tackle the problems of drug rights. (Photo: Cmdr. Roger Sultanousi)

According to a lawmakers 16-30 majority, the bill has the support of animal rights groups and equal representation. The bill would allow adult humans from nearly all biological categories to get arrested if they haven't already committed crimes and other vulnerable individuals who could be facing significant financial penalties. Utah's enactur percent has only 40 percent of all parental consent for an adult; that's not as far as human children go, but nearly half of teenagers are aborted at 18 months of age and illegal background checks don't exist for any other concern.

"Now focus on the serious crime aspect of the bill," spokeswoman Heather Chien said at the time. "Our believe hearing the issue of having the guy using the drug would be beneficial not just for the child, but the offender."

It comes as more and more farmers and livestock owners are scheduling change reviews. The Senate voted on Thursday to either let best-information request a vote on the pass or create a freedom of information request prioritized specifically by the state's enforcement motion, 54-12, to allow the vote by a vote of more than two to one.

NEWSLETTERS Get the AZ Memo newsletter delivered to your inbox We're sorry, but something went wrong Get the pulse of Arizona — Local news, in-depth state coverage and what it all means for you Please try again soon, or contact Customer Service at 1-800-332-6733. Delivery: Mon-Fri Invalid email address Thank you! You're almost signed up for AZ Memo Keep an eye out for an email to confirm your newsletter registration. More newsletters

In answer to a question posed by friend, lobbyist, and political activist Ed Priderout, this bill ultimately failed.

Though it remains to be seen what will happen in the future, a resolution Friday would call for the use of fewer than 5 percent of all adult diapers in a single year for the state's menial and surgical courier services.

The measure was also a tamper-resistant rabble fruit without trans fats.

"Despite some objections we need to make the safety mechanism work in a future we don't believe is most effective, and that's just who they are as I've been trying to get our labor. We have to use a combination of dog tags and earplugs, so that we can improve our conditions," said Rep. David Cale-Dero.

Copyright 2015 The Salt Lake Tribune.¡—endoftext—¿CALGARY — It was the two-day drive north through the republic – Ontario and the provinces – over the past three months that signs giving freemen extraordinary tourism in the province appear to have slipped away, leaving many of the seasonal magnetes at a bottom.

Really?

Rogue tards are wasting time in Canadians and anxieties are getting better.

Conspiracists have become "emerged," as experts characterize them, in a new sense, either as congenial naïvetés – quasi-religious adventurous sluts – rather than numb, territorial isolates like northern Canadians who are already born in the same spot, or simply as less flexible and maneuvering players that can deliver on their hard economic-mindedly defined tenets. Certain politicians are reading the Pearls Bible and happy that they lack cable, satellite TV or otherwise space-time worthy support in the province's often futile re-election campaigns. Yet many Canadians see the Alberta premier as more philosophical than conservative.

Some have joined the "unreliable" view that there is another side to the tale.

Amplification over the provincial election left Coyote Falls and Tire Centre, Conservative and Labor alike, shared the view a month ago on the periphery of the serenity of the Alberta electorate that the drums were driving on the Alberta government's provincial campaign. But the NDP MLA was also shaking that opinion with equal dismay when Ford's demand that union representation be halted by the federal government for the next two years yielded a somewhat sympathetic "no." And say many of the former leaders want more safeguards in the build-up of unions. One, minister Clifton Sanderson, acknowledged his former colleagues had brought with them new rules, regulations and so on that could be in their path. But, it remains a matter of interpretation.

"We welcome that a promise of transparency and clarity – that

Figure 12: Unconditional samples generated by $\gamma$-Hybrid [$\gamma = 0.01$, $\tau_{\text{Slide}}^{\omega=d/4}$, ALIGNED] trained on **OWT** dataset.

White House chief of staff Reince Priebus traveled to Washington, D.C., on Wednesday to be briefed by Vice President Mike Pence and other top advisers about the nature of the crisis in Ukraine and its potential impact on the United States. The new official, who was not authorized to speak publicly by the White House, met with senior American Russian policy experts and other world leaders as part of a trip to Moscow to meet with Ukrainian leader Mykola Azarov. The trip comes as tensions peak over President Donald Trump's handling of the refugee crisis and its potential for conflict with Russia. Jared Kushner's efforts to pump up the Dubrovnik deal have raised concerns that it could backfire. Kushner Trump is accused of colluding with the Kremlin, according to a New York Times report .

White House officials said that press secretary Sean Spicer called Kushner a "little stranger" when asked if he was familiar with the concerns swirling around the discussions, according to an account published by NBC News .

"The president-elect agreed to be briefed by my team on Russian meddling in our election process," Press Secretary Sean Spicer told NBC News in a telephone call, adding that he didn't have an authority to disclose details of the discussions. Versions of the meeting have drawn criticism from U.S. politicians, military officials and human rights advocates.

In a separate report, former acting FBI Director Andrew McCabe said he was not aware of any conversations with any Russian officials. He has said he did not know why the advisers were so closely involved in this scheduling conflict.

"I use very preliminary, telephone calls with colleagues both within our own department and overseas, and I have not had conversations with any of them," McCabe said of the meetings, which included senior officials and a White House official. The meetings involve joint efforts by the White House and individual individuals with the understanding that such a meeting would not occur, McCabe said, adding that no other member of the senior staff was involved.

"Please, contact the people you know with whom you have the knowledge in this room. Basically, I am asking you to do the work of reaching out to each one of the top officials associated with your Department of State," McCabe said.

"Since this is tense, I have trying to contact both the President and his staff and conduct the interview," McCabe said. Guidance

White House advisers held a series of meetings with top officials of the Department of State and its federal counterparts. However, the White House's meetings were themselves influenced by events to help the new administration rule out greater involvement in an escalating crisis that has killed more than 200,000 lives since last August.

In speeches delivered to the nation, Trump touted Friday's meeting on June 20 as another reminder of the importance of new sanctions against Russia. Trump said Russia's nuclear weapons "are insulting weak countries" and hoped "for a new normal" after a nosedive in the U.S. elections.

"We have a great relationship with Russia, and we continue to look forward to strengthening our relationship with them," said Trump.

The "warm" Russia policy has deepened the already long-standing divisions over the Obama administration, and Putin's recent ally Russia, who built a buffer zone along the Black Sea in 2013, ordered an invasion in 2014 of eastern Ukraine. White House spokesman Sean Spicer said he was "careful" with ties with Russia.¡—endoftext—¿On Tuesday night, LeBron James received news he was going — so often — from the Hall of Fame. Cleveland's most respected individual, who most known for his love of basketball, was getting a heart-to-heart with one of the most prestigious prospects in pro basketball history, sitting in the stands as the Knicks' first guard of the year. The official photo for the Knicks' current superstar, George Conte, was taken by the longtime reality television legend Wade Boggs.

We demand James get a heart. Exactly what he deserves is hard to guess. People to the point a man taken to heart who never weeks ago was the kind of person a champion of truth would want to meet or make close friends with without his inner purpose. After all, people feel very similar if they haven't seen Wade Boggs' latest film, Clump, in which he takes matters into his own hands and seizes his wife — the same wife that never had a child up for adoption.

This event has the potential to be a sour revenge for the Kevin Durant coronation that began a week ago – a winner's assuredly standard season that had little impact on James' looming future. But that's not the case here, either; the only one he is saying is that the uninspiring

Figure 13: Unconditional samples generated by $\gamma$-Hybrid [$\gamma = 0.01$, $\tau_{\text{Block}}^{\omega=d/64}$, SHIFTED] trained on **OWT** dataset.

be violent might be protected for one of the reasons he challenged.

Tsherberg is talking about former Republican Rep. Joe Burns who, in many ways tried to do better politically, canceled my appearance in 2012. She said that when the Republican president and we were playing with one other's decisions, it was the Chechen deal.

"Every time I think, there's nothing in the issue," she said. "There comes a time when I faced with reality in a very difficult way. It's not something I want to put the side of the my team past me."

During O'Reilly Factor in December 2011, when I dealt with most of my viewers, I watched the Obama Britain show again. "I was watching very, very closely, knowing that there was an opportunity, I knew I didn't want to be able to be able to help the Americans. I couldn't help that," I said.

Cack Obama also said several people without any authority, just due diligence, have commented on his plan to return to Chechnistan and Russia. T. Obama said he's not against Putin, and referred to himself as Putin-PM, who said he's taking "America's values forward."

"I think that's a turning point," Iza said in 2008 just before the start of my comments, referring to Iza's parent party, Moscow's Communist Party, in 1997, according to The Associated Press.

Asked whether Iza believes the economy and its security should run well into the Chechen agreement, and the leader ISIS was this month, Iza said that Trump is a far to go, more conservative agenda.

"He can't be a politician — he's not a tough guy who doesn't take a person," she said. "My people have respect too. My children are not civil andRoll but they do not [acc believe me]."

After speaking at Trump's Palm Beach, a bus stop in Texas on Sunday, his attorney, Republican Frank Agoodz, said neither he nor Iza accused me of arguing at the "Maxby Restaurant" in Texas.

"Williams, I want them to help, I don't want them to be able to help me," Iza was recorded as saying.

"He said, 'You have an opportunity in doing this and then you have to do this, you can do this; it's starting, that's getting the right people to do it, and he said, 'They're going and did, yeah,,'" she tells The Associated Press on Friday evening.

Sokherberg had not prior known Iza's release — since his public hearing, he hasn't has been expecting anything since then.

But Iza said this is a situation that will cause "Great pain" for all his family.

"It's important to my well-being, to care for others, about peace and just to be strong," she said. "As long as I don't, I won't. I think if I won't, I will have no apologies."

Their representatives before they gather on Saturday.

"It is a man with the very strong side of things, that has a long way to go," she said. "He's been singled out by the President. He has turned even more extreme off the map and is being led by the Republican party and the economy that are making a debt-laden economy out of his dream.

"The public is a mind-set, and his decision to speak to the public was a mind-set."

In a CNN video excerpt that includes her closing remarks, the then-U.S. Senate wasift to safety to stand in front of Republican House-time House Speaker and former White House special counsel Paul Pro Romanoff.¡—endoftext—¿New Villa in the midst of not making a cause in Sunday's clash at Villa, who started toestern to the title last season.

With Villa attacking you no certainly doubt it, if it was the second title they had won in the most recent five years, over the likes of Marc Canado, Robbie, Fernandes and CFOi.

But Villa think an answer to that battle over Canado is the help of the Vy's and in speeding up the relationship with the Portuguese forward in ways he hasn't shown in his past season.

West Villa turned their attention to defence in with Canado, who represented the England national team even last season after he became the first ever to score four goals.

He went on the same defensive side though, is now holding up front to rival one of the top midfielders and attack in the second half.

West Villa's second-half form last season however, has been done again in favour to substitute Scott-Fidane Jr, who later moved to Paris City on Wednesday.

John Gomez and João Aguero have moved to

Figure 14: Unconditional samples generated by $\epsilon$-Hybrid [$\epsilon = 0.01$, $\tau_{\text{Flat}}$, ALIGNED] trained on **OWT** dataset.

## G.2 Conditional Samples

The cat sat on the mat and looked up, staring at the ceiling as if something interesting were up there. It made no sound, but occasionally flicked its tail back and forth.

"Is he always like this?" I asked.

She smiled and nodded. "Pretty much. Sometimes he stares out the window too."

The cat glanced at me, seemed to consider something deeply philosophical for a moment, and then resumed staring upward.

"Do you think he sees things we don't?" I asked, half-joking.

"Probably," she replied, laughing softly. "Cats always seem to have a foot in another world, don't they?"

I chuckled, taking a sip from my coffee. "Maybe we should take notes."

"Maybe we should," she said, still smiling. "We might learn something."

The cat yawned lazily, stretched, and settled even more comfortably onto the mat, clearly deciding that whatever secrets the universe held could wait a little longer.

Figure 15: Conditional (conditioned on first 6 tokens) samples generated by $\epsilon$-Hybrid [$\epsilon = 0.01$, $\tau_{\text{Flat}}$, ALIGNED] trained on **OWT** dataset.

