# OpenReview forum: "Unifying Autoregressive and Diffusion-Based Sequence Generation"
_colmweb.org/COLM/2025/Conference — COLM 2025_

### Official Review · Reviewer_AaDn · 2025-05-05

**Rating:** 7
**Confidence:** 3
**Ethics Flag:** 1

**Summary:**

This paper shows how auto-regressive and diffusion-based methods for sequence generation can be unified conceptually through the notion of hyperschedules. In addition to the conceptual contribution, the paper introduces two hybrid noising processes, which interpolate between absorbing and uniform processes, enabling the model to fix past mistakes. Experiments show that the hybrid processes, together with a novel inference algorithm, can achieve lower perplexity than state-of-the-art diffusion-based models (while still not in par with auto-regressive models).

**Questions To Authors:**

I would like to know more about the evaluation of quality-diversity trade-offs, in particular whether you assume that perplexity is a measure of quality, since there are several studies showing that, while perplexity is a good measure of language modeling performance, it is a poor measure of text generation quality. See, for example, Carlsson et al. (2024) Branch-GAN: Improving Text Generation with (not so) Large Language Models. ICLR 2024.

Typos and such:
- Page 2: abstracting-out -> abstracting out
- Page 3 Yang et al. (2023) -> (Yang et al., 2023)

**Reasons To Accept:**

- Novel conceptual framework unifying auto-regressive and diffusion-based sequence generation.
- Technical innovations in noising processes and inference algorithms.
- Promising experimental results.

**Reasons To Reject:**

- I see no obvious reason to reject the paper, but I think the evaluation of quality-diversity trade-offs could be explained better.

---

> ### Author Response · Authors · 2025-05-30
>
> We thank the reviewer for acknowledging the novelty of our contributions, as well as raising an important point and providing typo fixes.
>
> > I would like to know more about the evaluation of quality-diversity trade-offs [...]
>
> We acknowledge that perplexity, while a common metric for language models, doesn't always align with human perception of sample quality. We include it for consistency with prior diffusion baselines, but recognize its limitations.
>
> To provide a more comprehensive evaluation, we complement perplexity with the MAUVE score. MAUVE assesses the divergence frontier between generated and human text in embedding space and, critically, has been shown to have a strong correlation with human judgments of both text quality and diversity. As depicted in **Figure 4 (right)** in the paper, our hybrid models consistently achieve a superior perplexity-MAUVE Pareto frontier compared to all baselines. Furthermore, the qualitative samples in **Appendix G** demonstrate that these improvements in perplexity and MAUVE translate into more coherent and varied outputs.
>
> The camera ready will clarify the limitations of perplexity, and that combining it with MAUVE provides a more balanced evaluation framework.
>
> **Action point for camera ready**: Discuss the limitations of perplexity and why we include MAUVE evaluations.
>
> [1] Pillutela et al., MAUVE: Measuring the Gap Between Neural Text and Human Text using Divergence Frontiers

---

> > ### Comment · Reviewer_AaDn · 2025-06-04
> > **Thanks**
> >
> > Thanks for your clarification.

---

### Official Review · Reviewer_XPkA · 2025-05-12

**Rating:** 7
**Confidence:** 4
**Ethics Flag:** 1

**Summary:**

The paper introduces two key innovations: hyper-schedules and hybrid token-wise noising processes. Hyper-schedules assign distinct noise schedules to individual token positions, generalizing both AR models (e.g., GPT) and conventional diffusion models (e.g., SEDD, MDLM) as special cases. Hybrid noising processes interpolate between absorbing and uniform processes, enabling models to correct past errors. Additionally, the paper proposes a novel inference algorithm, Adaptive Correction Sampler (ACS), which leverages the features of hybrid processes. To support efficient training and inference, the authors design attention masks compatible with KV-caching. The methods achieve state-of-the-art perplexity and generate diverse, high-quality sequences across standard benchmarks, demonstrating a promising path for autoregressive diffusion-based sequence generation.

**Reasons To Accept:**

I think overall this paper is theoretically sound.  It reveals the fundamental connection between autoregressive and diffusion models and provides new insights for sequence generation model research and expands the design space for models.
The introduction of hyper-schedules and hybrid noising processes offers new technical approaches for sequence generation. Hyper-schedules enable fine-grained control over noise schedules for different token positions, while hybrid noising processes combine the advantages of absorbing and uniform processes, enhancing model performance and flexibility.

**Reasons To Reject:**

I think the missing part is the analysis on the training and inference overhead. Although the paper proposes attention masks compatible with KV-caching to improve efficiency, the training and inference processes of the model remain relatively complex and resource-intensive. So I would like to see more results and explanations in this direction.

---

> ### Author Response · Authors · 2025-05-30
>
> We would like to thank the reviewer for their thoughtful feedback and for recognizing the theoretical contributions of our work.
>
> > I think the missing part is the analysis on the training [...]
>
> Regarding **training overhead**, please refer to our detailed response to Reviewer NmzT: as shown in Appendix B.1, our two-stage pretraining and fine-tuning schedule requires **no more time** than other diffusion-based LMs of comparable size (110 M parameters).
>
> For **inference overhead**, we leverage our hyperschedule custom attention masks with KV-caching. Appendix F.7 (Table 11) demonstrates that our latency is **lower** than MDLM’s fastest caching strategy. Furthermore, Tables 7, 9, and 10 show how varying the generation rate **ρ** (tokens generated per network call) can **further accelerate** inference—an emergent capability of diffusion-based generators that lets users trade off speed and quality on the fly.
>
> For unconditional generation:
>
> - the total number of times that the model is called is $O(d/\rho + \omega)$ (where $d$ is the sequence length, $\rho$ is the hyperschedule's token generation rate and $\omega$ is its window width);
> - **without** KV-caching, the total number of tokens to be processed (including re-processings of already-known tokens) is $O(d^2/\rho)$; and
> - **with** KV-caching, the total number of tokens to be processed is: $O(\omega d/\rho)$.
>
> **Action point for camera ready**: Add inference complexity analysis to Appendix B.4.
>
> Finally, we wish to emphasize that our contribution introduces many different models and variations which, we concede, may seem complex altogether. However, each specific instantiation of our framework is much simpler.

---

> > ### Comment · Reviewer_XPkA · 2025-06-06
> >
> > Thanks for address my concern, i will raise my score accordingly.

---

### Official Review · Reviewer_NmzT · 2025-05-13

**Rating:** 7
**Confidence:** 3
**Ethics Flag:** 1

**Summary:**

This paper unifies autoregressive and diffusion-based sequence generation by introducing hyperschedules, which assign different noise levels to each token position, encompassing both paradigms as special cases. It proposes hybrid noising processes that blend absorbing and uniform noise to allow correction of past mistakes during generation. There is an adaptive correction sampler, that further enhances output quality. The approach supports efficient inference via KV-caching and achieves good results on multiple benchmarks, narrowing the gap between diffusion and autoregressive models.

**Questions To Authors:**

1. You design attention masks to support KV-caching for efficient inference, but the training cost of your hybrid models is not discussed. Could you provide more details about this?
2.  All experiments use relatively small transformer models (110M). Have you attempted scaling your method to larger models (e.g., 1B+)? If not, what are the expected challenges or limitations when applying at that scale?
3. The ACS introduces a new hyperparameter η. How sensitive is model performance to the choice of η? Is ACS stable across different datasets and tasks, or does it require careful tuning for each setup?

**Reasons To Accept:**

1. The paper proposed a novel formulation that unifies autoregressive and diffusion-based sequence generation, which may contribute a fresh perspective.
2. The proposed hybrid approach interpolates between absorbing and uniform noising. It allows the model to correct past token errors during generation.
3. The empirical results are strong
4. The experiments and analysis is well-structured.

**Reasons To Reject:**

1. Lack of training cost analysis compared to AR models and diffusion models
2. The experiments are mainly done on small models. I'm not sure when we scale up the mode, whether the advantage of this method will still be there. (I understand if the authors don't have resources to train models)
3. For hybrid noising part, the theoretical justification is kind of limited.
4. Lack of some related works, for example, The Matrix: Infinite-Horizon World Generation with Real-Time Moving Control, which is related to this work.

---

> ### Author Response · Authors · 2025-05-30
>
> We thank the reviewer for acknowledging the importance of our contribution, as well as for their questions, which should help us clarify our manuscript.
>
> > You design attention masks to support KV-caching for efficient inference, but the training cost of your hybrid models is not discussed. Could you provide more details about this?
>
> Our hybrid models follow a two-stage training approach, detailed in Appendix B.1. Initially, we pretrain models using the $\tau\_{flat}$ hyperschedule: approximately 14 days on 8×A100-80GB GPUs for OpenWebText (OWT) and 7 days on 2×A100-80GB GPUs for LM1B. We then fine-tune for each configuration over an additional 150K steps -- ~3 days (8×A100 GPUs) on OWT and ~36 hours (2×A100 GPUs) on LM1B. This two stage approach helps us get as many runs as possible with our limited compute budget and also proves that the hyperschedule variants are robust enough to improve performance while being limited to the late-stage fine-tuning.
>
> The table below compares training throughput across different models (single A100-80GB GPU, no custom kernels, largest batch size divisible by 2; 64 for most methods and 32 for “Ours Efficient” due to its processing strategy). It highlights the efficiency gains achieved by our method through the use of customized attention masks. Here, we set the window size as ω = d/4; using smaller values of ω further amplifies the efficiency benefits of our attention masks during training.
>
>
> | Method                  | Throughput (token/ms) | Effective Throughput (active token/ms) |
> |-------------------------|------------------------|--------------------------------|
> | Ours Non-Efficient (ω = d/4)     | 100.31                 | 25.07                          |
> | Ours Efficient (ω = d/4)         | 88.04                  | 44.02                          |
> | MDLM  [1]                  | 97.64                  | 54.42                          |
> | SEDD [2]                   | 99.29                  | 56.01                          |
> | AR                      | 127.65                 | 127.65                         |
>
>
> By leveraging our efficient attention masks and doubling context length through concatenation (clean and noisy sequences), we achieve nearly twice the active token throughput compared to our non-efficient training, thus significantly reducing overall training cost. This confirms that our efficient training paradigm allows our hyperschedule variants to achieve an effective throughput comparable to other diffusion baselines.
>
> **Action point for camera ready**: Add efficiency throughput table and discussion to the appendix.
>
> [1] Sahoo et al., Simple and Effective Masked Diffusion Language Models
>
> [2] Lou et al., Discrete Diffusion Modeling by Estimating the Ratios of the Data Distribution
>
>
> > All experiments use relatively small transformer models (110M). Have you attempted scaling your method to larger models (e.g., 1B+)? If not, what are the expected challenges or limitations when applying at that scale?
>
>
> While fully scaling to 1B+ models lies beyond this paper’s scope, we are actively exploring this direction: early experiments  (intended for future work) on a 350M–parameter prototype confirm that our hybrid models follow the expected, near-linear scaling trends in perplexity and per-token training time [3]. To further cut training time at scale, we’re integrating flex-attention [4] custom kernels, which dramatically reduce the overhead of our windowed attention masks. Moreover, recent work [5] has successfully scaled diffusion-based language models up to 8B parameters without introducing fundamental architectural bottlenecks, reinforcing our belief that our method remains practical and may even further benefit from larger-scale deployment.
>
> **Action point for camera ready**: Add a short discussion about scaling to appendix.
>
> [3] Nie et. al,  Scaling up Masked Diffusion Models on Text.
>
> [4] https://pytorch.org/blog/flexattention/
>
> [5] Nie et. al, Large Language Diffusion Models.
>
> (Continued below.)

---

> > ### Author Response · Authors · 2025-05-30
> >
> > (Continued from above.)
> >
> > > The ACS introduces a new hyperparameter η. How sensitive is model performance to the choice of η? Is ACS stable across different datasets and tasks, or does it require careful tuning for each setup?
> >
> >
> > | Sampler         | MAUVE (ρ=8) | MAUVE (ρ=4) | MAUVE (ρ=2) | MAUVE (ρ=1) | Gen PPL (ρ=8) | Gen PPL (ρ=4) | Gen PPL (ρ=2) | Gen PPL (ρ=1) | Entropy (ρ=8) | Entropy (ρ=4) | Entropy (ρ=2) | Entropy (ρ=1) |
> > |-----------------|--------------|--------------|--------------|--------------|----------------|----------------|----------------|----------------|----------------|----------------|----------------|----------------|
> > | Original Sampler | 0.950       | 0.944       | 0.848       | 0.779       | 130.78        | 124.75        | 121.90        | 129.52        | 5.51          | 5.47          | 5.49          | 5.50          |
> > | ACS (η=0.25)     | 0.955       | 0.821       | 0.859       | 0.928       | 79.64         | 65.06         | 55.05         | 49.09         | 5.35          | 5.28          | 5.24          | 5.19          |
> > | ACS (η=0.05)     | 0.846       | 0.990       | 0.865       | 0.936       | 105.94        | 93.91         | 83.83         | 77.16         | 5.46          | 5.48          | 5.31          | 5.29          |
> > | ACS (η=0.01)     | 0.848       | 0.928       | 0.957       | 0.947       | 84.28         | 69.84         | 61.35         | 43.98         | 5.36          | 5.33          | 5.28          | 5.18          |
> > | ACS (η=0.001)    | 0.871       | 0.949       | 0.919       | 0.998       | 80.37         | 64.42         | 55.48         | 45.80         | 5.35          | 5.31          | 5.25          | 5.15          |
> > | ACS (η=0.5)      | 0.740       | 0.987       | 0.859       | 0.837       | 31.86         | 64.42         | 55.48         | 45.80         | 5.35          | 5.31          | 5.25          | 5.15          |
> > | ACS (η=0.75)     | 0.883       | 0.770       | 0.999       | 0.293       | 28.51         | 24.57         | 23.49         | 22.23         | 5.03          | 4.90          | 4.79          | 4.56          |
> > | ACS (η=0.99)     | 0.618       | 0.582       | 0.754       | 0.508       | 28.05         | 24.94         | 22.01         | 21.26         | 4.98          | 4.86          | 4.66          | 4.41          |
> >
> > We address the sensitivity of the Adaptive Correction Sampler (ACS) to the hyperparameter η in Appendix F.6 of the paper. To further support this analysis, we have included additional results (shown in the table above) for a wider range of η values, including higher values such as 0.5, 0.75, and 0.99. These evaluations were performed on a single model family: the ε-Hybrid [ε=0.01, τ_flat, ALIGNED].
> >
> > Across the board, our findings confirm that ACS is generally stable and robust with respect to η. In the recommended range of 0.001 < η < 0.25, the model achieves a favorable trade-off between generation quality (low Gen PPL), diversity (high Entropy), and distributional alignment with human text (high MAUVE).
> >
> > The additional values make clear that larger η values (e.g., 0.75 and 0.99) lead to overcorrection, resulting in degraded sample quality—visible in sharp drops in MAUVE and Entropy, particularly at low sampling steps (ρ). Though Gen PPL continues to decrease in those cases, the generated text becomes less diverse and less natural, indicating that these improvements in perplexity come at the cost of generation fidelity.
> >
> > We also observe that η = 0.5 is somewhat borderline: while it still achieves competitive PPL and moderate diversity, it does not improve over the recommended range and introduces unnecessary aggressiveness in correction.
> >
> > **Action point for camera ready**: Extend Table 10 in Appendix F.6.
> >
> >
> > > For hybrid noising part, the theoretical justification is kind of limited.
> >
> > Our answers to Reviewer WiDd provides additional details about the theoretical foundations that may be of interest to reviewer NmzT.
> >
> > If the question concerned "Why hybrid?", our main justifications are that "absorb" ("mask") empirically performs better than "uniform", but the backward process of "absorb" cannot "fix" past mistakes (whereas "uniform" can). Our "hybrid" process gets the best of both worlds. Ultimately, our justification is empirical.
> >
> >
> > > Lack of some related works, for example, The Matrix: Infinite-Horizon World Generation with Real-Time Moving Control, which is related to this work.
> >
> > We thank the reviewer for this related work. We are not yet familiar with this work, but at a first glance the Shift-Window Denoising Process Model (SwinDPM) indeed looks sufficiently related to be cited in our introduction.
> >
> > **Action point for camera ready**: Read SwinDPM and cite it.
> >
> > Again, we thank the reviewer for their valuable input.

---

> > ### Comment · Reviewer_NmzT · 2025-06-05
> >
> > Thanks for the clarification, i will keep the score

---

### Official Review · Reviewer_WiDd · 2025-05-22

**Rating:** 4
**Confidence:** 4
**Ethics Flag:** 1

**Summary:**

This paper proposes noise schedules (hyperschedules) that
* interpolate AR and Masked Diffusion Models (MDMs).
* interpolate Uniform State and Masked Diffusion Models.
In doing so, they achieve competitive performance on language modeling benchmarks.

**Reasons To Accept:**

Although papers like Block Diffusion [1] interpolate between autoregressive models (AR) and masked discrete diffusion models (MDMs), combining uniform state diffusion models (USDMs) with MDMs is a novel contribution. A key limitation of MDMs is that they cannot revise their predictions during the reverse sampling process. Integrating USDMs with MDMs addresses this limitation, enabling iterative correction and making it a significant advancement for the community.




[1] Arriola et al., 2025 "Block Diffusion: Interpolating Between Autoregressive and Diffusion Language Models"

**Reasons To Reject:**

The main reason I’m currently inclined to reject this paper is the lack of a principled ELBO formulation for their hyperschedules. It appears that the authors evaluate their models using the training objectives described in Appendix `B.2`. As a result, the perplexity (PPL) values reported in `Tables 1 and 2` may be incorrect or misleading.


Deriving the ELBO for their hyperschedules should be straightforward. Specifically, the authors would need to use `Equation 10` of [2], and:
* Reformulate their state transition matrices as rate matrices as per `Appendix C.1` in [1], and
* Reformulate the score function using the mean parameterization approach, as outlined in `Appendix C.5`.


**I would be happy to increase my score to 7** if the authors are able to adequately address this concern.

[1] Sahoo et al., 2024 "Simple and Effective Masked Diffusion Language Models"

[2] Lou et al., 2024 "Discrete Diffusion Modeling by Estimating the Ratios of the Data Distribution"

---

> ### Author Response · Authors · 2025-05-30
>
> We thank the reviewer for acknowledging the novelty of our work, as well as for providing valuable inputs that allowed us to identify specific action points that should improve the overall clarity of our contribution.
>
> > It appears that the authors evaluate their models using the training objectives described in Appendix B.2. As a result, the perplexity (PPL) values reported in Tables 1 and 2 may be incorrect or misleading.
>
> We first wish to clarify that the training objectives described in Appendix B.2, as well as the hyperschedule and other details of the underlying curriculum (masking and/or uniform noise), *are not directly involved in our "Test PPL" metric*. We now realize that this may not be clear in the present state of our Appendix C.1.
>
> All Test PPL scores computed by us, including both $\gamma$-Hybrid and $\epsilon$-Hybrid, as well as baseline diffusion models (unless mentioned "reported in..."), are all evaluated with the same code implementing Algorithm 1, using the loss listed in Appendix C.1 (i.e., **not** the one described in Appendix B.2). This algorithm estimates an upper bound for the perplexity, reflecting how good the **score/logit-predicting neural network** -- **not** the "full inference strategy" -- is at "language modeling" (in the sense of "predicting the probability for the masked tokens"). The inference-time specificities of the model could actually make it effectively better at language modelling than this metric captures.
>
> The training objectives described in Appendix B.2, the choice of hyperschedule and the value of $\epsilon$ or $\gamma$, can **indirectly** affect the Test PPL metric through their impact on the training of the neural network's weights. The camera ready will clarify these points.
>
> **Action point for camera ready**: Add clarifications about the "Test Perplexity" metric.
>
> (Continued below.)

---

> > ### Author Response · Authors · 2025-05-30
> >
> > (Continued from above.)
> >
> > > lack of a principled ELBO formulation for their hyperschedules
> >
> > Despite it not being required to justify our perplexity evaluation, we agree that providing a more detailed consideration of the losses has its own interest. To simplify comparison with [1] and [2], we reinterpret our hyperschedules as position-specific schedules $\sigma\_{t}^{i} = \sigma\_{\tau\_{T-t}^{i}}$, $\bar{\sigma}\_{t}^{i} = \bar{\sigma}\_{\tau\_{T-t}^{i}}$ and $\alpha\_{t}^{i} = \alpha\_{\tau\_{T-t}^{i}}$, thus aligning our "direction of time" with these two works.
> >
> > For $\gamma$-Hybrid, we first notice that Equation 10 in [2] holds for arbitrary diffusion matrix $Q\_{t}$ (which in [2] represents the "rates" mentioned by the reviewer; contrast with [1] which uses the notation "$Q\_{t}$" for what we here call the "evolution operator"). As mentioned in the beginning of Section 3.3 in [2], such a general $Q\_{t}$ would be exponential in size, i.e., we would have a $\mathbb{R}^{d \lvert \mathcal{Y} \rvert \times d \lvert \mathcal{Y} \rvert}$ stochastic matrix to specify for each $t$. Instead, they consider (Equation (13)) $d$ independent subspaces each independently subjected to $\sigma\_{t} Q^{\text{tok}} \in \mathbb{R}^{d \times d}$. Let's write $\left[ Q^{\text{tok}} \right]\_{i} \in \mathbb{R}^{d \lvert \mathcal{Y} \rvert \times d \lvert \mathcal{Y} \rvert}$ the effect of $Q^{\text{tok}}$ on the $i$-th token. Notice that $\left[ Q^{\text{tok}} \right]\_{i} \left[Q^{\text{tok}} \right]\_{j} = \left[Q^{\text{tok}}\right]\_{j} \left[Q^{\text{tok}}\right]\_{i}$ because the actions on the $i$-th and $j$-th tokens are independent for $i \neq j$: these operators commute. [2]'s full diffusion matrix is thus $Q\_{t} = \sum\_{i=1}^{d} \sigma\_{t} \left[Q^{\text{tok}}\right]\_{i}$, with solution $\prod\_{i=1}^{d} \left[\exp(\bar{\sigma}\_{t} Q^{\text{tok}})\right]\_{i}$ for the evolution operator. **In our $\gamma$-Hybrid case**, the hyperschedule amounts to using a different schedule $\sigma\_{t}^{i}$ for different tokens $i$: our full diffusion matrix is $Q\_{t} = \sum\_{i=1}^d \sigma\_{t}^{i} \left[Q^{\text{tok}}\right]\_{i}$, resulting in the evolution operator $\prod\_{i=1}^{d} \left[\exp(\bar{\sigma}\_{t}^{i} Q^{\text{tok}})\right]\_{i}$. If Equation 10 holds in [2], then it also holds for our hyperschedule (provided that we substitute the appropriate position-specific schedules in the implementation). Finally, the evolution operator for our specific choice of $Q^{\text{tok}}$ is already discussed in our Appendix A.
> >
> > For $\epsilon$-Hybrid, we directly specify the (per token) evolution operator on line 172 of our manuscript. First note that, in the special case $\epsilon=0$, we fall back on the above case with $\gamma = 0$ by defining $\sigma\_{t}^{i}$ such that $\alpha\_{t}^{i} = \exp(-\bar{\sigma}\_{t}^{i})$: **our hyperschedule *per se* is compatible with MDLM** as per the same argument as before. However, this argument does not directy apply for $\epsilon > 0$, as there is no finite $Q\_{t}$ that may capture the discrete jump induced by the operator $\mathbb{I} + \epsilon Q\_{\text{Uniform}}$ as soon as nonzero noise applies. One possibility is to "stretch" this discrete jump into a continuous transition over a period of $\ln(1-\epsilon)$ time units (by comparing line 172 with expressions in Appendix A). For simplicity, we insert this time period "before time zero", using
> > $$
> > Q\_{t} = \begin{cases}
> > \sum\_{i=1}^{d} \left[ Q\_{\text{Uniform}} \right]\_{i} & \text{for $t < 0$} \\\\
> > \sum\_{i=1}^{d} \sigma\_{t}^{i} \left[ Q\_{\text{Absorb}} \right]\_{i} & \text{for $t \ge 0$}
> > \end{cases}
> > $$
> > as the diffusion matrix (with $\sigma\_{t}^{i}$ such that $\alpha\_{t}^{i} = \exp(-\bar{\sigma}\_{t}^{i})$). Using $y$ (instead of $x\_{0}$) for the unperturbed data and $x'$ (instead of $y$) for the quantity being summed over, Equation (10) from [2] becomes
> > $$
> > \int\_{\ln (1-\epsilon)}^{T} \mathbb{E}\_{x\_t \sim p\_{t|0}(\cdot|y)} \sum\_{x' \neq x\_{t}} Q\_{t}(x\_{t}, x') \left( s\_{\theta}(x\_{t},t)\_{x'} - \frac{p\_{t|0}(x'|y)}{p\_{t|0}(x\_{t}|y)} \ln s\_{\theta}(x\_{t},t)\_{x'} + K\left( \frac{p\_{t|0}(x'|y)}{p\_{t|0}(x\_t|y)} \right) \right) \textup{d}t
> > $$
> > It would be interesting to see if training with such a "two stages" approach could yield better models. Similarly, one could potentially derive from first principles something akin to our ACS. However, we deem these questions outside the scope of the present work.
> >
> >
> > We sincerely thank the reviewer for triggering these reflections. A cleaned-up version of the above will be added to the appendix of an eventual camera-ready.
> >
> > **Action point for camera ready**: Add clarifications as to the theoretical backing and derivation of our training loss functions.
> >
> > [1] Sahoo et al., 2024 "Simple and Effective Masked Diffusion Language Models"
> >
> > [2] Lou et al., 2024 "Discrete Diffusion Modeling by Estimating the Ratios of the Data Distribution"

---

> ### Author Response · Authors · 2025-06-09
>
> Dear Reviewer WiDd,
>
> We wish to respectfully remind you that the author-reviewer discussion period is scheduled to end tomorrow, June 10th. This is the final opportunity for us to address any further questions you may have.
>
> Thank you for your time and consideration of our paper.
>
> Sincerely,
>
> The Authors

---

### Author Response · Authors · 2025-05-30
**Summary of proposed changes**

We sincerely thank all reviewers for acknowledging the value of our work and providing isightful points to our attention. We list below diffent improvements made possible by those reviews, which we're commiting to include in an eventual camera-ready version.

- Add clarifications about the "Test Perplexity" metric. (WiDd)
- Add clarifications as to the theoretical backing and derivation of our training loss functions. (WiDd)
- Add efficiency throughput table and discussion to the appendix. (NmzT)
- Add a short discussion about scaling to appendix. (NmzT)
- Extend Table 10 in Appendix F.6. (NmzT)
- Read SwinDPM and cite it. (NmzT)
- Add inference complexity analysis to Appendix B.4. (XPkA)
- Discuss the limitations of perplexity and why we include MAUVE evaluations. (AaDn)

We welcome further discussions with the reviewers.

---

### Decision · Program_Chairs · 2025-07-08

**Decision:**

Accept

**Comment:**

This paper has universally strong reviews. Reviewer WiDd brought up concerns on fair test comparisons, and would revise the score 7 to if resolved. I believe their concerns were resolved by the author rebuttal and appendix C.1 of the original paper.

A couple comments that do not affect evaluation:
1. Nit: Position-dependent schedule would be a more precise term than hyperschedule. Hyperschedule does sound better.
2. Question: I am curious what a product process would look like. For example, taking the product of uniform diffusion as well as masking via the addition of a per-position settled bit. Perhaps this would be more useful with learned hyperschedules.
3. Nit: The set of variables is larger than necessary. For example, $y$ is not necessary as latent diffusion is not considered in this paper. In general, it's nice to minimize notation.